# Conformalized matrix completion

**Yu Gui    Rina Foygel Barber    Cong Ma**

Department of Statistics, University of Chicago
{yugui,rina,congm}@uchicago.edu

## Abstract

Matrix completion aims to estimate missing entries in a data matrix, using the assumption of a low-complexity structure (e.g., low rank) so that imputation is possible. While many effective estimation algorithms exist in the literature, uncertainty quantification for this problem has proved to be challenging, and existing methods are extremely sensitive to model misspecification. In this work, we propose a distribution-free method for predictive inference in the matrix completion problem. Our method adapts the framework of conformal prediction, which provides confidence intervals with guaranteed distribution-free validity in the setting of regression, to the problem of matrix completion. Our resulting method, conformalized matrix completion (`cmc`), offers provable predictive coverage regardless of the accuracy of the low-rank model. Empirical results on simulated and real data demonstrate that `cmc` is robust to model misspecification while matching the performance of existing model-based methods when the model is correct.

## 1 Introduction

Matrix completion, the task of filling in missing entries in a large data matrix, has a wide range of applications such as gene-disease association analysis in bioinformatics (Natarajan and Dhillon, 2014), collaborative filtering in recommender systems (Rennie and Srebro, 2005), and panel data prediction and inference in econometrics (Amjad et al., 2018; Bai and Ng, 2021; Athey et al., 2021). If the underlying signal is assumed to be low-rank, a range of estimation algorithms have been proposed in the literature, including approaches based on convex relaxations of rank (Candes and Plan, 2010; Candès and Tao, 2010; Koltchinskii et al., 2011; Foygel and Srebro, 2011; Negahban and Wainwright, 2012; Yang and Ma, 2022) and nonconvex optimization over the space of low-rank matrices (Keshavan et al., 2010; Jain et al., 2013; Sun and Luo, 2016; Ma et al., 2020).

While the problem of estimation has been explored through many different approaches in the literature, the question of uncertainty quantity for matrix completion remains challenging and under-explored. Cai et al. (2016) and Carpentier et al. (2018) construct confidence intervals for the missing entries using order-wise concentration inequalities, which could be extremely loose. Chen et al. (2019) proposes a de-biased estimator for matrix completion and characterizes its asymptotic distribution, which then yields entrywise confidence intervals are then derived for the underlying incoherent matrix with i.i.d. Gaussian noise. See also Xia and Yuan (2021); Farias et al. (2022) for related approaches.

The effectiveness of the aforementioned uncertainty quantification methods rely heavily on the exact low-rank structure as well as assumptions on the tail of the noise. In practice, the low-rank assumption can only be met approximately, and heavy-tailed noise is quite common in areas such as macroeconomics and genomics (Nair et al., 2022). Without exact model specifications, the solution to the matrix completion algorithm is hard to interpret and the asymptotic distribution of the obtained estimator is no longer valid. The dependence of model-based approaches on exact model assumptions motivates us to answer the question: *can we construct valid and short confidence intervals for the*

37th Conference on Neural Information Processing Systems (NeurIPS 2023).

*missing entries that are free of both model assumptions and of constraints on which estimation algorithms we may use?*

## 1.1 Conformal prediction for distribution-free uncertainty quantification

Our proposed method is built on the idea of conformal prediction. The conformal prediction framework, developed by Vovk et al. (2005) and Shafer and Vovk (2008), has drawn significant attention in recent years in that it enables the construction of distribution-free confidence intervals that are valid with exchangeable data from any underlying distribution and with any "black-box" algorithm. To reduce computation in full conformal prediction, variants exist based on data splitting (Vovk et al., 2005; Lei et al., 2018), leave-one-out (Barber et al., 2021), cross-validation (Vovk, 2015; Barber et al., 2021), etc.

When distribution shift is present, with covariate shift as a special case, the exchangeability is violated, and Tibshirani et al. (2019) proposes a more general procedure called weighted conformal prediction that guarantees the validity under the weighted exchangeability of data. Barber et al. (2022) further relaxes the exchangeability assumption on the data and characterizes the robustness of a generalized weighted approach via an upper bound for the coverage gap. Other related works (Lei and Candès, 2020; Candès et al., 2021; Jin et al., 2023) adapt the conformal framework to applications in causal inference, survival analysis, etc., and study the robustness of weighted conformal prediction with estimated weights.

## 1.2 Main contributions

In this paper, we adapt the framework of conformal prediction for the problem of matrix completion, and make the following contributions:

1. We construct distribution-free confidence intervals for the missing entries in matrix completion via conformal prediction. The validity is free of any assumption on the underlying matrix and holds regardless of the choice of estimation algorithms practitioners use. To achieve this, we prove the (weighted) exchangeability of unobserved and observed units when they are sampled (possibly nonuniformly) without replacement from a finite population.

2. When the sampling mechanism is unknown, we develop a provable lower bound for the coverage rate which degrades gracefully as the estimation error of the sampling probability increases.

3. In addition, to improve computational efficiency when faced with a large number of missing entries, we propose a one-shot conformalized matrix completion approach that only computes the weighted quantile once for all missing entries.

## 2 Problem setup

In this section, we formulate the matrix completion problem and contrast our distribution-free setting with the model-based settings that are more common in the existing literature. For a partially-observed $d_1 \times d_2$ matrix, the subset $\mathcal{S} \subseteq [d_1] \times [d_2]$ denotes the set of indices where data is observed—that is, our observations consist of $(M_{ij})_{(i,j) \in \mathcal{S}}$. In much of the matrix completion literature, it is common to assume a signal-plus-noise model for the observations,

$$M_{ij} = M_{ij}^* + E_{ij},$$

where $\mathbf{M}^* = (M_{ij}^*)_{(i,j) \in [d_1] \times [d_2]}$ is the "true" underlying signal while $(E_{ij})_{(i,j) \in \mathcal{S}}$ denotes the (typically zero-mean) noise. Frequently, it is assumed that $\mathbf{M}^*$ is low-rank, or follows some other latent low-dimensional structure, so that recovering $\mathbf{M}^*$ from the observed data is an identifiable problem. In works of this type, the goal is to construct an estimator $\widehat{\mathbf{M}}$ that accurately recovers the underlying $\mathbf{M}^*$. In contrast, in a more model-free setting, we may no longer wish to hypothesize a signal-plus-noise model, and can instead assume that we are observing a random subset of entries of a *deterministic* matrix $\mathbf{M}$; in this setting, estimating $\mathbf{M}$ itself becomes the goal.

For both frameworks, many existing results focus on the problem of *estimation* (either of $\mathbf{M}^*$ or of $\mathbf{M}$), with results establishing bounds on estimation errors under various assumptions. For instance, see Candes and Plan (2010); Candès and Tao (2010); Negahban and Wainwright (2012); Chen et al. (2020) for results on estimating $\mathbf{M}^*$ (under stronger conditions, e.g., a low-rank and incoherent signal,

and zero-mean sub-Gaussian noise), or Srebro and Shraibman (2005); Foygel and Srebro (2011) for results on estimating $\mathbf{M}$ (under milder conditions); note that in the latter case, it is common to instead refer to *predicting* the entries of $\mathbf{M}$, e.g., temperature or stock market returns, since we often think of these as noisy observations of some underlying signal. On the other hand, relatively little is known about the problem of *uncertainty quantification* for these estimates: given some estimator $\widehat{\mathbf{M}}$, *can we produce a confidence interval around each $\widehat{M}_{ij}$ that is (asymptotically) guaranteed to contain the target with some minimum probability?*

The results of Chen et al. (2019) address this question under strong assumptions, namely, a low-rank and incoherent signal $\mathbf{M}^*$, plus i.i.d. Gaussian noise $(E_{ij})_{(i,j)\in[d_1]\times[d_2]}$. Zhao and Udell (2020) proposes another inferential procedure under the Gaussian copula assumption for the data. In this work, we provide a complementary answer that instead addresses the model-free setting: without relying on assumptions, we aim to produce a provably valid confidence interval for the entries $M_{ij}$ of $\mathbf{M}$ (not of $\mathbf{M}^*$, because without assuming a low-rank model, there is no meaningful notion of an "underlying signal").

To do so, we will treat the matrix $\mathbf{M}$ as deterministic, while the randomness arises purely from the random subset of entries that are observed. More specifically, we assume

$$Z_{ij} = \mathbb{1}\{(i,j) \text{ is observed}\} \sim \text{Bern}(p_{ij}), \text{ independently for all } (i,j) \in [d_1] \times [d_2], \quad (1)$$

and let $\mathcal{S} = \{(i,j) \in [d_1] \times [d_2] : Z_{ij} = 1\}$ be the subset of observed locations. We consider the setting where the sample probabilities $p_{ij}$'s are nonzero. After observing $\mathbf{M}_{\mathcal{S}} = (M_{ij})_{(i,j)\in\mathcal{S}}$, our goal is to provide confidence intervals $\{\widehat{C}(i,j)\}_{(i,j)\in\mathcal{S}^c}$ for all unobserved entries, with $1 - \alpha$ coverage over the unobserved portion of the matrix—that is, we would like our confidence intervals to satisfy

$$\mathbb{E}\left[\text{AvgCov}(\widehat{C}; \mathbf{M}, \mathcal{S})\right] \geq 1 - \alpha, \quad (2)$$

where

$$\text{AvgCov}(\widehat{C}; \mathbf{M}, \mathcal{S}) = \frac{1}{|\mathcal{S}^c|} \sum_{(i,j)\in\mathcal{S}^c} \mathbb{1}\left\{M_{ij} \in \widehat{C}(i,j)\right\} \quad (3)$$

measures the "average coverage rate" of the confidence intervals.[1] The goal of this work is to provide an algorithm for constructing confidence intervals $\{\widehat{C}(i,j)\}_{(i,j)\in\mathcal{S}^c}$ based on the observed entries $\mathbf{M}_{\mathcal{S}}$, satisfying (2) with no assumptions on $\mathbf{M}$. In particular, given the strong success of various matrix completion algorithms for the problem of estimation, we would like to be able to provide uncertainty quantification around any base estimator—i.e., given any algorithm for producing an estimate $\widehat{\mathbf{M}}$, our method constructs confidence intervals $\{\widehat{C}(i,j)\}_{(i,j)\in\mathcal{S}^c}$ around this initial estimate.

**Notation.** For a matrix $\mathbf{A} = (A_{ij}) \in \mathbb{R}^{m_1 \times m_2}$, we define $\|\mathbf{A}\|_\infty := \max_{(i,j)\in[m_1]\times[m_2]} |A_{ij}|$, and $\|\mathbf{A}\|_1 := \sum_{(i,j)\in[m_1]\times[m_2]} |A_{ij}|$. We use $\mathbf{A}_{i,\cdot}$ and $\mathbf{A}_{\cdot,j}$ to refer to the $i$th row and $j$th column of $\mathbf{A}$, respectively. For any $t \in \mathbb{R}$, $\delta_t$ denotes the distribution given by a point mass at $t$.

## 3 Conformalized matrix completion

We next present our method, conformalized matrix completion (cmc). Our procedure adapts the split conformal prediction method (Vovk et al., 2005) to the problem at hand. As is standard in the conformal prediction framework, the goal of cmc is to provide uncertainty quantification (i.e., confidence intervals) around the output of any existing estimation procedure that the analyst chooses to use—this is a core strength of the conformal methodology, as it allows the analyst to use state-of-the-art estimation procedures without compromising on the validity of inference. At a high level, the cmc procedure will output a confidence interval of the form

$$\widehat{C}(i,j) = \widehat{M}_{ij} \pm \widehat{q} \cdot \widehat{s}_{ij}$$

for the matrix value $M_{ij}$ at each unobserved entry $(i,j)$. Here, after splitting the observed entries into a training set and a calibration set, the training set is used to produce $\widehat{M}_{ij}$ (a point estimate for the

---

[1] To handle the degenerate case where $\mathcal{S} = [d_1] \times [d_2]$ (i.e., we happen to have observed the entire matrix) and thus $\mathcal{S}^c = \varnothing$, we simply define $\text{AvgCov}(\widehat{C}; \mathbf{M}, \mathcal{S}) \equiv 1$.

target value $M_{ij}$) and $\widehat{s}_{ij}$ (a scaling parameter that estimates our uncertainty for this point estimate); then, the calibration set is used to tune the scalar $\widehat{q}$, to adjust the width of these confidence intervals and ensure coverage at the desired level $1 - \alpha$.[2]

---

**Algorithm 1** Conformalized matrix completion (`cmc`)

---

1: **Input**: target coverage level $1 - \alpha$; data splitting proportion $q \in (0, 1)$; observed entries $\mathbf{M}_{\mathcal{S}}$.
2: Split the data: draw $W_{ij} \overset{\text{i.i.d.}}{\sim} \text{Bern}(q)$, and define training and calibration sets

$$\mathcal{S}_{\text{tr}} = \{(i, j) \in \mathcal{S} : W_{ij} = 1\}, \quad \text{and} \quad \mathcal{S}_{\text{cal}} = \{(i, j) \in \mathcal{S} : W_{ij} = 0\}.$$

3: Using the training data $\mathbf{M}_{\mathcal{S}_{\text{tr}}}$ indexed by $\mathcal{S}_{\text{tr}} \subseteq [d_1] \times [d_2]$, compute:
   - An initial estimate $\widehat{\mathbf{M}}$ using any matrix completion algorithm (with $\widehat{M}_{ij}$ estimating the target $M_{ij}$);
   - Optionally, a local uncertainty estimate $\widehat{\mathbf{s}}$ (with $\widehat{s}_{ij}$ estimating our relative uncertainty in the estimate $\widehat{M}_{ij}$), or otherwise set $\widehat{s}_{ij} \equiv 1$;
   - An estimate $\widehat{\mathbf{P}}$ of the observation probabilities (with $\widehat{p}_{ij}$ estimating $p_{ij}$, the probability of entry $(i, j)$ being observed).

4: Compute normalized residuals on the calibration set, $R_{ij} = \frac{|M_{ij} - \widehat{M}_{ij}|}{\widehat{s}_{ij}}$, $(i, j) \in \mathcal{S}_{\text{cal}}$.
5: Compute estimated odds ratios for the calibration set and test set, $\widehat{h}_{ij} = \frac{1 - \widehat{p}_{ij}}{\widehat{p}_{ij}}$, $(i, j) \in \mathcal{S}_{\text{cal}} \cup \mathcal{S}^c$, and then compute weights for the calibration set and test point,

$$\widehat{w}_{ij} = \frac{\widehat{h}_{ij}}{\displaystyle\sum_{(i', j') \in \mathcal{S}_{\text{cal}}} \widehat{h}_{i'j'} + \max_{(i', j') \in \mathcal{S}^c} \widehat{h}_{i'j'}}, \; (i, j) \in \mathcal{S}_{\text{cal}}, \; \widehat{w}_{\text{test}} = \frac{\displaystyle\max_{(i, j) \in \mathcal{S}^c} \widehat{h}_{ij}}{\displaystyle\sum_{(i', j') \in \mathcal{S}_{\text{cal}}} \widehat{h}_{i'j'} + \max_{(i', j') \in \mathcal{S}^c} \widehat{h}_{i'j'}}. \quad (4)$$

6: Compute threshold $\widehat{q} = \text{Quantile}_{1-\alpha}\left(\sum_{(i,j) \in \mathcal{S}_{\text{cal}}} \widehat{w}_{ij} \cdot \delta_{R_{ij}} + \widehat{w}_{\text{test}} \cdot \delta_{+\infty}\right)$.
7: **Output**: confidence intervals $\widehat{C}(i, j) = \widehat{M}_{ij} \pm \widehat{q} \cdot \widehat{s}_{ij}$ for each unobserved entry $(i, j) \in \mathcal{S}^c$.

---

### 3.1 Exchangeability and weighted exchangeability

We split the set of observed indices $\mathcal{S}$ into a training and a calibration set $\mathcal{S} = \mathcal{S}_{\text{tr}} \cup \mathcal{S}_{\text{cal}}$ as is shown in Algorithm 1. We should notice that the sampling without replacement may introduce implicit dependence as well as a distribution shift from the i.i.d. sampling. Thus the two sets are dependent, which is different from the split conformal methods in regression problems. Before we present the coverage guarantees for our method, we first examine the role of (weighted) exchangeability in this method, to build intuition for how the method is constructed.

#### 3.1.1 Intuition: the uniform sampling case

First, for intuition, consider the simple case where the entries are sampled with constant probability, $p_{ij} \equiv p$. If this is the case, then the set of calibration samples and the test point are exchangeable—that is, for $(i_*, j_*)$ denoting the test point that is drawn uniformly from $\mathcal{S}^c$, if we are told that the combined set $\mathcal{S}_{\text{cal}} \cup \{(i_*, j_*)\}$ is equal to $\{(i_1, j_1), \ldots, (i_{n_{\text{cal}}+1}, j_{n_{\text{cal}}+1})\}$ in no particular order (where $n_{\text{cal}} = |\mathcal{S}_{\text{cal}}|$), then the test point location $(i_*, j_*)$ is *equally likely* to be at any one of these $n_{\text{cal}} + 1$ locations. Consequently, by exchangeability, the calibration residuals $\{R_{ij} : (i, j) \in \mathcal{S}_{\text{cal}}\}$, defined in Algorithm 1 above, are exchangeable with the test residual $R_{i_* j_*} = |M_{i_* j_*} - \widehat{M}_{i_* j_*}| / \widehat{s}_{i_* j_*}$; as a result, we can construct our confidence interval $\widehat{C}(i_*, j_*)$ with $\widehat{q}$ determined by the $(1 - \alpha)$-quantile of the calibration residuals $\{R_{ij} : (i, j) \in \mathcal{S}_{\text{cal}}\}$.

---

[2]More generally, `cmc` can be implemented with any choice of a *nonconformity score* that determines the shape of the resulting confidence intervals—and indeed, this generalization allows for categorical rather than real-valued matrix entries $M_{ij}$. We will address this extension in the Supplementary Material, as well as a version of `cmc` based on *full conformal* rather than split conformal, in order to avoid data splitting.

Indeed, this is exactly what cmc does in this case: if we are aware that sampling is uniform, it would naturally follow that we estimate $\widehat{p}_{ij} \equiv \widehat{p}_0$ by some (potentially data-dependent) scalar $\widehat{p}_0$; consequently, all the weights are given by $\widehat{w}_{ij} \equiv \frac{1}{n_{\mathrm{cal}}+1}$ and $\widehat{w}_{\mathrm{test}} = \frac{1}{n_{\mathrm{cal}}+1}$, and thus $\widehat{q}$ is the (unweighted) $(1-\alpha)$-quantile of the calibration residuals (with a small correction term, i.e., the term $+\widehat{w}_{\mathrm{test}} \cdot \delta_{+\infty}$ appearing in the definition of $\widehat{q}$, to ensure the correct probability of coverage at finite sample sizes).

### 3.1.2 Weighted exchangeability for the matrix completion problem

Next, we move to the general case, where now the sampling may be highly nonuniform. First suppose the sampling probabilities $p_{ij}$ are known, and suppose again that we are told that the combined set $\mathcal{S}_{\mathrm{cal}} \cup \{(i_*, j_*)\}$ is equal to $\{(i_1, j_1), \ldots, (i_{n_{\mathrm{cal}}+1}, j_{n_{\mathrm{cal}}+1})\}$ in no particular order. In this case, the test point $(i_*, j_*)$ is *not* equally likely to be at any one of these $n_{\mathrm{cal}} + 1$ locations. Instead, we have the following result:

**Lemma 3.1.** *Following the notation defined above, if $(i_*, j_*) \mid \mathcal{S} \sim \mathrm{Unif}(\mathcal{S}^c)$, it holds that*

$$\mathbb{P}\big\{(i_*, j_*) = (i_k, j_k) \mid \mathcal{S}_{\mathrm{cal}} \cup \{(i_*, j_*)\} = \{(i_1, j_1), \ldots, (i_{n_{\mathrm{cal}}+1}, j_{n_{\mathrm{cal}}+1})\}, \mathcal{S}_{\mathrm{tr}}\big\} = w_{i_k j_k},$$

*where we define the weights* $w_{i_k j_k} = \frac{h_{i_k j_k}}{\sum_{k'=1}^{n_{\mathrm{cal}}+1} h_{i_{k'} j_{k'}}}$ *for odds ratios given by* $h_{ij} = \frac{1-p_{ij}}{p_{ij}}$.

Consequently, while the test point's residual $R_{i_* j_*} = |M_{i_* j_*} - \widehat{M}_{i_* j_*}| \big/ \widehat{s}_{i_* j_*}$ is not exchangeable with the calibration set residuals $\{R_{ij} : (i,j) \in \mathcal{S}_{\mathrm{cal}}\}$, the framework of weighted exchangeability (Tibshirani et al., 2019) tells us that we can view $R_{i_* j_*}$ as a draw from the *weighted* distribution that places weight $w_{i_k j_k}$ on each residual $R_{i_k j_k}$—and in particular,

$$\mathbb{P}\big\{M_{i_* j_*} \in \widehat{M}_{i_* j_*} \pm q^*_{i_*, j_*} \cdot \widehat{s}_{i_* j_*}\big\} = \mathbb{P}\big\{R_{i_* j_*} \leq q^*_{i_*, j_*}\big\} \geq 1-\alpha,$$

where $q^*_{i_*, j_*} = \mathrm{Quantile}_{1-\alpha}\big(\sum_{k=1}^{n_{\mathrm{cal}}+1} w_{i_k j_k} \cdot \delta_{R_{i_k j_k}}\big)$. Noting that this threshold $q^*_{i_*, j_*}$ may depend on the test location $(i_*, j_*)$, to accelerate the computation of prediction sets for all missing entries, we instead propose a one-shot weighted conformal approach by upper bounding the threshold uniformly over all test points: $q^*_{i_*, j_*} \leq q^* := \mathrm{Quantile}_{1-\alpha}\big(\sum_{(i,j) \in \mathcal{S}_{\mathrm{cal}}} w^*_{ij} \cdot \delta_{R_{ij}} + w^*_{\mathrm{test}} \cdot \delta_{+\infty}\big)$, for $(i,j) \in \mathcal{S}_{\mathrm{cal}}$,

$$w^*_{ij} = \frac{h_{ij}}{\displaystyle\sum_{(i',j') \in \mathcal{S}_{\mathrm{cal}}} h_{i'j'} + \max_{(i',j') \in \mathcal{S}^c} h_{i'j'}}, \quad w^*_{\mathrm{test}} = \frac{\displaystyle\max_{(i,j) \in \mathcal{S}^c} h_{ij}}{\displaystyle\sum_{(i',j') \in \mathcal{S}_{\mathrm{cal}}} h_{i'j'} + \max_{(i',j') \in \mathcal{S}^c} h_{i'j'}}. \tag{5}$$

Now the threshold $q^*$ no longer depends on the location $(i_*, j_*)$ of the test point[3]. If the true probabilities $p_{ij}$ were known, then, we could define "oracle" conformal confidence intervals

$$\widehat{C}^*(i,j) = \widehat{M}_{ij} \pm q^* \cdot \widehat{s}_{ij}, \quad (i,j) \in \mathcal{S}^c.$$

As we will see in the proofs below, weighted exchangeability ensures that, if we do have oracle knowledge of the $p_{ij}$'s, then these confidence intervals satisfy the goal (2) of $1-\alpha$ average coverage. Since the $p_{ij}$'s are not known in general, our algorithm simply replaces the oracle weights $\boldsymbol{w}^*$ in (5) with their estimates $\widehat{\boldsymbol{w}}$ defined in (4), using $\widehat{p}_{ij}$ as a plug-in estimate of the unknown $p_{ij}$.

### 3.2 Theoretical guarantee

In the setting where the $p_{ij}$'s are not known but instead are estimated, our cmc procedure simply repeats the procedure described above but with weights $\widehat{\boldsymbol{w}}$ calculated using $\widehat{p}_{ij}$'s in place of $p_{ij}$'s. Our theoretical results therefore need to account for the errors in our estimates $\widehat{p}_{ij}$, to quantify the coverage gap of conformal prediction with the presence of these estimation errors. To do this, we define an estimation gap term

$$\Delta = \frac{1}{2} \sum_{(i,j) \in \mathcal{S}_{\mathrm{cal}} \cup \{(i_*, j_*)\}} \left| \frac{\widehat{h}_{ij}}{\sum_{(i',j') \in \mathcal{S}_{\mathrm{cal}} \cup \{(i_*, j_*)\}} \widehat{h}_{i'j'}} - \frac{h_{ij}}{\sum_{(i',j') \in \mathcal{S}_{\mathrm{cal}} \cup \{(i_*, j_*)\}} h_{i'j'}} \right|, \tag{6}$$

---

[3]Comparison between the one-shot weighted approach and the exact weighted approach (Algorithm 3) is conducted in Section B.1

where $(i_*, j_*)$ denotes the test point. Effectively, $\Delta$ is quantifying the difference between the "oracle" weights, $w_{ij}^*$, defined in (5), and the estimated weights, $\widehat{w}_{ij}$, defined in (4), except that $\Delta$ is defined relative to a specific test point, while the weights $\widehat{w}$ (and $w^*$, for the oracle version) provide a "one-shot" procedure that is universal across all possible test points, and is thus slightly more conservative.

**Theorem 3.2.** *Let $\widehat{\mathbf{M}}$, $\widehat{s}$, and $\widehat{\mathbf{P}}$ be estimates constructed using any algorithms, which depend on the data only through the training samples $\mathbf{M}_{\mathcal{S}_{\mathrm{tr}}}$ at locations $\mathcal{S}_{\mathrm{tr}}$. Then, under the notation and definitions above, conformalized matrix completion (`cmc`) satisfies*

$$\mathbb{E}\big[\mathtt{AvgCov}(\widehat{C}; \mathbf{M}, \mathcal{S})\big] \geq 1 - \alpha - \mathbb{E}[\Delta].$$

In the homogeneous case, where we are aware that sampling is uniform (i.e., $p_{ij} \equiv p$ for some potentially unknown $p \in (0,1)$), our error in estimating the weights is given by $\Delta \equiv 0$ (since $h_{ij}$ is constant across all $(i,j)$, and same for $\widehat{h}_{ij}$—and therefore, the true and estimated weights are all equal to $\frac{1}{n_{\mathrm{cal}}+1}$). In this setting, therefore, we would achieve the exact coverage goal (2), with $\mathbb{E}\big[\mathtt{AvgCov}(\widehat{C}; \mathbf{M}, \mathcal{S})\big]$ guaranteed to be at least $1 - \alpha$.

### 3.3 Examples for missingness

To characterize the coverage gap explicitly, we present the following concrete examples for $\mathbf{P}$ and show how the coverage gap $\Delta$ can be controlled. Technical details are shown in the Supplementary Material.

#### 3.3.1 Logistic missingness

Suppose that the missingness follows a logistic model, with $\log(p_{ij}/(1-p_{ij})) = -\log(h_{ij}) = u_i + v_j$ for some $\boldsymbol{u} \in \mathbb{R}^{d_1}$ and $\boldsymbol{v} \in \mathbb{R}^{d_2}$, where we assume $\boldsymbol{u}^\top \mathbf{1} = 0$ for identifiability. This model is closely related to logistic regression with a diverging number of covariates (Portnoy, 1988; He and Shao, 2000; Wang, 2011). Following Chen et al. (2023), we estimate $\boldsymbol{u}, \boldsymbol{v}$ via maximum likelihood,

$$\widehat{\boldsymbol{u}}, \widehat{\boldsymbol{v}} = \arg\max_{\boldsymbol{u}, \boldsymbol{v}} \mathcal{L}(\boldsymbol{u}, \boldsymbol{v}) \qquad \text{subject to} \quad \mathbf{1}^\top \boldsymbol{u} = 0.$$

Here the log-likelihood is defined by

$$\mathcal{L}(\boldsymbol{u}, \boldsymbol{v}) = \sum_{(i,j) \in [d_1] \times [d_2]} \left\{ -\log\left(1 + \exp(-u_i - v_j)\right) + \mathbf{1}_{(i,j) \in \mathcal{S}_{\mathrm{tr}}^c} \log\left(1 - q + \exp(-u_i - v_j)\right) \right\}.$$

Consequently, we define the estimate of $p_{ij}$ as $\widehat{p}_{ij} = (1 + \exp(-\widehat{u}_i - \widehat{v}_j))^{-1}$, for which one can show that $\mathbb{E}[\Delta] \lesssim \sqrt{\frac{\log(\max\{d_1, d_2\})}{\min\{d_1, d_2\}}}$. The proof of this upper bound is shown in Section A.3.1.

#### 3.3.2 Missingness with a general link function

More broadly, we can consider a general link function, where we assume $\log(p_{ij}/(1-p_{ij})) = -\log(h_{ij}) = \phi(A_{ij})$, where $\phi$ is a link function and $\mathbf{A} \in \mathbb{R}^{d_1 \times d_2}$ is the model parameter. The logistic model we introduced above corresponds to the case when $\phi$ is the identity function and $\mathbf{A} = \boldsymbol{u}\mathbf{1}^\top + \mathbf{1}\boldsymbol{v}^\top$ is a rank-2 matrix. In this general setup, if $\mathrm{rank}(\mathbf{A}) = k^*$ and $\|\mathbf{A}\|_\infty \leq \tau$, we can apply one-bit matrix completion (Davenport et al., 2014) to estimate $\{p_{ij}\}$. More precisely, we define the log-likelihood

$$\mathcal{L}_{\mathcal{S}_{\mathrm{tr}}}(\mathbf{B}) = \sum_{(i,j) \in \mathcal{S}_{\mathrm{tr}}} \log(\psi(B_{ij})) + \sum_{(i,j) \in \mathcal{S}_{\mathrm{tr}}^c} \log(1 - \psi(B_{ij})),$$

where $\psi(t) = q(1 + e^{-\phi(t)})^{-1}$ (here rescaling by the constant $q$ is necessary to account for subsampling the training set $\mathcal{S}_{\mathrm{tr}}$ from the observed data indices $\mathcal{S}$), and solve the following optimization problem

$$\widehat{\mathbf{A}} = \arg\max_{\mathbf{B}} \mathcal{L}_{\mathcal{S}_{\mathrm{tr}}}(\mathbf{B}) \qquad \text{subject to} \quad \|\mathbf{B}\|_* \leq \tau\sqrt{k^* d_1 d_2}, \ \ \|\mathbf{B}\|_\infty \leq \tau, \tag{7}$$

where $\|\mathbf{B}\|_*$ is the nuclear norm of $\mathbf{B}$. Consequently, we let $\widehat{h}_{ij} = \exp(-\phi(\widehat{A}_{ij}))$, which leads to $\mathbb{E}[\Delta] \lesssim \min\{d_1, d_2\}^{-1/4}$. The proof of this upper bound is shown in Section A.3.2.

# 4 Simulation studies

In this section, we conduct numerical experiments to verify the coverage guarantee of conformalized matrix completion (`cmc`) using both synthetic and real datasets. As the validity of `cmc` is independent of the choice of base algorithm, we choose alternating least squares (`als`) (Jain et al., 2013) as the base algorithm (results using a convex relaxation (Fazel et al., 2004; Candes and Recht, 2012; Chen et al., 2020) are shown in the Supplementary Material). We use `cmc-als` to refer to this combination. We use $\mathtt{AvgCov}(\widehat{C})$ (the average coverage, defined in (3)), and average confidence interval length, $\mathtt{AvgLength}(\widehat{C}) = \frac{1}{|\mathcal{S}^c|} \sum_{(i,j)\in\mathcal{S}^c} \mathrm{length}\big(\widehat{C}(i,j)\big)$, to evaluate the performance of different methods. All the results in this section can be replicated with the code available at `https://github.com/yugjerry/conf-mc`.

## 4.1 Synthetic dataset

Throughout the experiments, we set the true rank $r^* = 8$, the desired coverage rate $1 - \alpha = 0.90$, and report the average results over 100 random trials.

**Data generation.** In the synthetic setting, we generate the data matrix by $\mathbf{M} = \mathbf{M}^* + \mathbf{E}$, where $\mathbf{M}^*$ is a rank $r^*$ matrix, and $\mathbf{E}$ is a noise matrix (with distribution specified below). The low-rank component $\mathbf{M}^*$ is generated by $\mathbf{M}^* = \kappa \mathbf{U}^* \mathbf{V}^{*\top}$, where $\mathbf{U}^* \in \mathbb{R}^{d_1 \times r^*}$ is the orthonormal basis of a random $d_1 \times r^*$ matrix consisting of i.i.d. entries from a certain distribution $\mathcal{P}_{u,v}$ (specified below) and $\mathbf{V}^* \in \mathbb{R}^{d_2 \times r^*}$ is independently generated in the same manner. In addition, the constant $\kappa$ is chosen such that the average magnitude of the entries $|M_{ij}^*|$ is 2. Following the observation model (1), we only observe the entries in the random set $\mathcal{S}$.

**Implementation.** First, when implementing `als`, we adopt $r$ as the hypothesized rank of the underlying matrix. For model-based methods (Chen et al., 2019), by the asymptotic distributional characterization $\widehat{M}_{ij} - M_{ij}^* \overset{d}{\to} \mathcal{N}(0, \theta_{ij}^2)$ and $E_{ij} \perp (\widehat{M}_{ij} - M_{ij}^*)$, one has $\widehat{M}_{ij} - M_{ij} \overset{d}{\to} \mathcal{N}(0, \theta_{ij}^2 + \sigma^2)$. Consequently, the confidence interval for the model-based methods can be constructed as

$$\widehat{M}_{ij} \pm q_{1-\alpha/2}\sqrt{\widehat{\theta}_{ij}^2 + \widehat{\sigma}^2},$$

where $q_\beta$ is the $\beta$th quantile of $\mathcal{N}(0,1)$. The estimates are obtained via $\widehat{\sigma}^2 = \|\mathbf{M}_{\mathcal{S}_{\mathrm{tr}}} - \widehat{\mathbf{M}}_{\mathcal{S}_{\mathrm{tr}}}\|_{\mathrm{F}}^2/|\mathcal{S}_{\mathrm{tr}}|$ and $\widehat{\theta}_{ij}^2 = (\widehat{\sigma}^2/\widehat{p})(\|\widehat{\mathbf{U}}_{i,\cdot}\|^2 + \|\widehat{\mathbf{V}}_{j,\cdot}\|^2)$ with the SVD $\widehat{\mathbf{M}} = \widehat{\mathbf{U}}\widehat{\boldsymbol{\Sigma}}\widehat{\mathbf{V}}^\top$. For `cmc-als`, we obtain $\widehat{\mathbf{M}}, \widehat{s}$ from $\mathbf{M}_{\mathcal{S}_{\mathrm{tr}}}$ by running `als` on this training set (and taking $\widehat{s}_{ij} = (\widehat{\theta}_{ij}^2 + \widehat{\sigma}^2)^{1/2}$). The probabilities $\widehat{p}_{ij}$'s are estimated via $\widehat{p}_{ij} \equiv (d_1 d_2 q)^{-1}|\mathcal{S}_{\mathrm{tr}}|$ for this setting where the $p_{ij}$'s are homogeneous.

### 4.1.1 Homogeneous missingness

In the homogeneous case, we consider the following four settings with $d_1 = d_2 = 500$:

- **Setting 1: large sample size + Gaussian noise**. $p = 0.8$, $\mathcal{P}_{u,v} = \mathcal{N}(0,1)$, and $E_{ij} \sim \mathcal{N}(0,1)$.
- **Setting 2: small sample size + Gaussian noise**. $p = 0.2$, $\mathcal{P}_{u,v} = \mathcal{N}(0,1)$, and $E_{ij} \sim \mathcal{N}(0,1)$.
- **Setting 3: large sample size + heavy-tailed noise**. $p = 0.8$, $\mathcal{P}_{u,v} = \mathcal{N}(0,1)$, and $E_{ij} \sim 0.2\,\mathsf{t}_{1.2}$, where $t_\nu$ denotes the $t$ distribution with $\nu$ degrees of freedom.
- **Setting 4: violation of incoherence**. $p = 0.8$, $\mathcal{P}_{u,v} = t_{1.2}$, and $E_{ij} \sim \mathcal{N}(0,1)$.

Figure 1 displays the results (i.e., the coverage rate and the interval length) for both `cmc-als` and `als` when we vary the hypothesized rank $r$ from 2 to 40. The oracle length is the difference between the $(1 - \alpha/2)$-th and $(\alpha/2)$-th quantiles of the underlying distribution for $E_{ij}$. As can be seen, `cmc-als` achieves nearly exact coverage regardless of the choice of $r$ across all four settings. When the hypothesized rank $r$ is chosen well, `cmc-als` is often able to achieve an average length similar to that of the oracle.

For the model-based method `als`, when we underestimate the true rank ($r < r^*$), we may still see adequate coverage since $\widehat{\sigma}$ tends to over-estimate $\sigma$ by including the remaining $r^* - r$ factors as noise. However, if we overestimate the rank ($r > r^*$), then `als` tends to overfit the noise with additional factors and under-estimate $\sigma$ regardless of the choice of $r$, leading to significant undercoverage

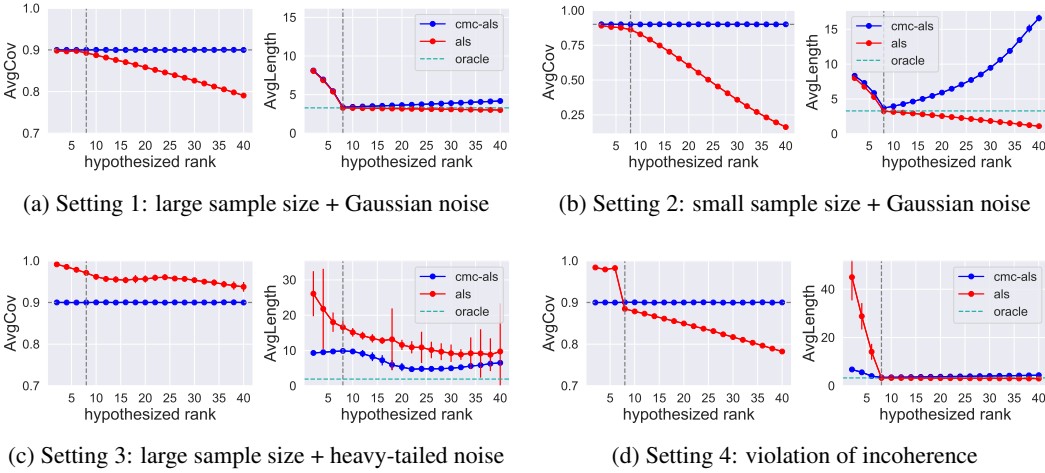

(a) Setting 1: large sample size + Gaussian noise

(b) Setting 2: small sample size + Gaussian noise

(c) Setting 3: large sample size + heavy-tailed noise

(d) Setting 4: violation of incoherence

Figure 1: Comparison between `cmc-als` and `als`.

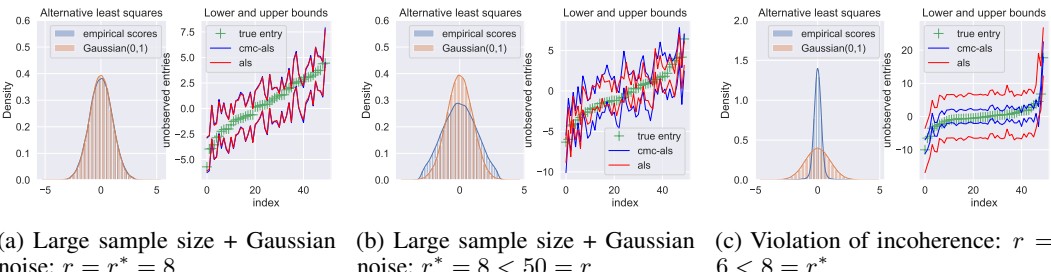

(a) Large sample size + Gaussian noise: $r = r^* = 8$

(b) Large sample size + Gaussian noise: $r^* = 8 < 50 = r$

(c) Violation of incoherence: $r = 6 < 8 = r^*$

Figure 2: Histogram of standardized scores for `als` and prediction lower and upper bounds for $50$ distinct unobserved entries.

in Settings 1, 2 and 4. In Setting 3 with heavy-tailed noise, `als` tends to be conservative due to overestimating $\sigma$. Overall, Figure 1 confirms our expectation that the validity of the model-based estimates relies crucially on a well specified model, and it fails to hold when sample size is small (cf. Figure 1b), when noise is heavy tailed (cf. Figure 1c), and when the underlying matrix is not incoherent (cf. Figure 1d).

In Figure 2, we present the histogram of standardized scores $(\widehat{M}_{ij} - M_{ij})/\sqrt{\widehat{\theta}_{ij}^2 + \widehat{\sigma}^2}$ and the plot of the upper and lower bounds for three settings. In Figure 2a, when the model assumptions are met and $r = r^*$, the scores match well with the standard Gaussian and the prediction bounds produced by `als` and `cmc-als` are similar. With the same data generating process, when the rank is overparametrized, the distribution of scores cannot be captured by the standard Gaussian, thus the quantiles are misspecified. As we can see from the confidence intervals, `als` tends to have smaller intervals which lead to the undercoverage. In the last setting, the underlying matrix is no longer incoherent. When the rank is underestimated, the $r^* - r$ factors will be captured by the noise term and the high heterogeneity in the entries will further lead to overestimated noise level. As a result, the intervals by `als` are much larger while the conformalized intervals are more adaptive to the magnitude of entries.

#### 4.1.2 Heterogeneous missingness

Now we move on to the case with heterogeneous missingness, where the observed entries are no longer sampled uniformly. To simulate this setting, we draw $\{a_{il} : i \leq d_1, l \leq k^*\}$ i.i.d. from $\mathrm{Unif}(0,1)$ and $\{b_{lj} : l \leq k^*, j \leq d_2\}$ i.i.d. from $\mathrm{Unif}(-0.5, 0.5)$, and define sampling probabilities by

$$\log(p_{ij}/(1 - p_{ij})) = \sum_{l=1}^{k^*} a_{il} b_{lj}.$$

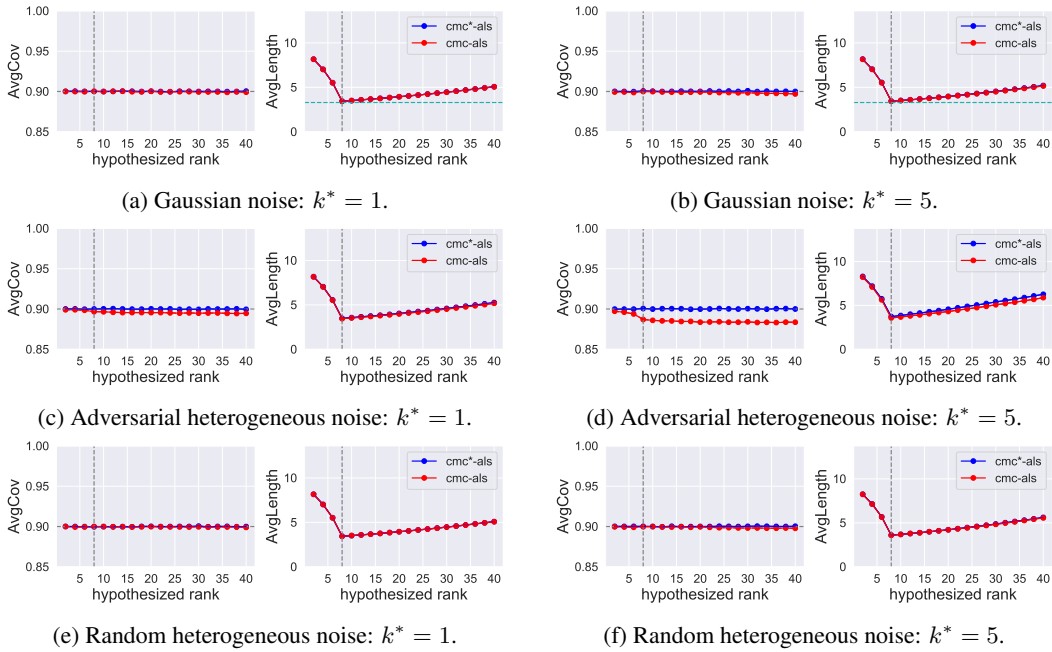

(a) Gaussian noise: $k^* = 1$.

(b) Gaussian noise: $k^* = 5$.

(c) Adversarial heterogeneous noise: $k^* = 1$.

(d) Adversarial heterogeneous noise: $k^* = 5$.

(e) Random heterogeneous noise: $k^* = 1$.

(f) Random heterogeneous noise: $k^* = 5$.

Figure 3: Comparison between `cmc` with true and estimated weights under heterogeneous missingness.

We consider two settings with $k^* = 1$ and $k^* = 5$, respectively. In both settings, $\widehat{p}_{ij}$ is constructed by estimating $\widehat{a}_{il}$ and $\widehat{b}_{lj}$ via constrained maximum likelihood estimation as is shown in Example 3.3.2.

To analyze the effect of estimation error predicted by Theorem 3.2, we consider three matrix generation processes for each choice of $k^*$, where $\mathbf{M}^*$ is generated in the same way as in Section 4.1.1 with $d_1 = d_2 = d = 500$, $\mathcal{P}_{u,v} = \mathcal{N}(0, 1)$:

- **Gaussian homogeneous noise**: $E_{ij} \sim \mathcal{N}(0, 1)$ independently.

- **Adversarial heterogeneous noise**: $E_{ij} \sim \mathcal{N}(0, \sigma_{ij}^2)$ independently, where we take $\sigma_{ij} = 1/2p_{ij}$. For this setting, note that the high-noise entries (i.e., those entries that are hardest to predict) occur in locations $(i, j)$ that are least likely to be observed during training.

- **Random heterogeneous noise**: $E_{ij} \sim \mathcal{N}(0, \sigma_{ij}^2)$ independently, where $(\sigma_{ij})_{(i,j) \in [d_1] \times [d_2]}$ is drawn from the same distribution as $(1/2p_{ij})_{(i,j) \in [d_1] \times [d_2]}$.

We write `cmc*-als` to denote the conformalized method (with `als` as the base algorithm) where we use oracle knowledge of the true $p_{ij}$'s for weighting, while `cmc-als` uses estimates $\widehat{p}_{ij}$. The results are shown in Figures 3, for the settings $k^* = 1$ and $k^* = 5$ (i.e., the rank of $\mathbf{P}$). Under both homogeneous noise and random heterogeneous noise, both `cmc*-als` and `cmc-als` achieve the correct coverage level, with nearly identical interval length. For adversarial heterogeneous noise, on the other hand, `cmc*-als` achieves the correct coverage level as guaranteed by the theory, but `cmc-als` shows some undercoverage due to the errors in the estimates $\widehat{p}_{ij}$ (since now the variance of the noise is adversarially aligned with these errors); nonetheless, the undercoverage is mild. In Section B.2 in the appendix, the local coverage of `cmc-als` conditioning $p_{ij} = p_0$ for varying $p_0$ is presented.

## 4.2 Real data application

We also compare conformalized approaches with model-based approaches using the Rossmann sales dataset[4] (Farias et al., 2022). This real data provides the underlying fixed matrix $\mathbf{M}$; the missingness pattern is generated artificially, as detailed below. We focus on daily sales (the unit is 1K dollar) of 1115 drug stores on workdays from Jan 1, 2013 to July 31, 2015 and hence the underlying matrix has dimension $1115 \times 780$. The hypothesized rank $r$ is varied from 5 to 60 with step size 5 and we set

---

[4] `https://www.kaggle.com/c/rossmann-store-sales`

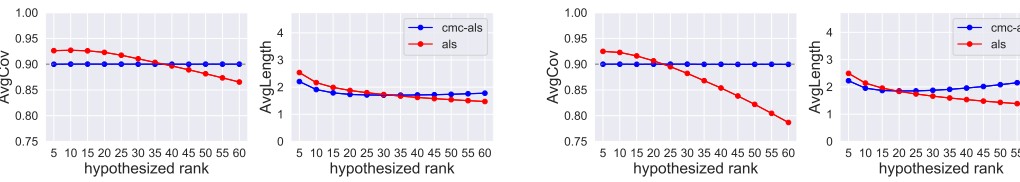

(a) `als` vs `cmc-als`: homogeneous missingness.

(b) `als` vs `cmc-als`: logistic missingness.

Figure 4: Comparison between conformalized and model-based matrix completion with sales dataset.

the target level to be $1 - \alpha = 0.9$. We apply random masking for 100 independent trials and report the average for `AvgCov` and `AvgLength` in Figure 4.

- **Homogeneous** $p_{ij}$. For each entry, we first apply random masking $1 - Z_{ij} \sim \mathrm{Bern}(0.2)$ independently. In Figure 4a, conformalized approach has exact coverage at $1 - \alpha$ but `als` tends to be conservative when $r$ is small and loses coverage when $r$ increases to 50.

- **Heterogeneous** $p_{ij}$. After drawing $p_{ij}$'s from $\log(p_{ij}/(1 - p_{ij})) = -\log(h_{ij}) = \phi(A_{ij})$ in Section 4.1.2 with $k^* = 1$, $\widehat{\mathbf{P}}$ is obtained via the constrained maximum likelihood estimator in Example 3.3.2. In Figure 4b, `cmc-als` has coverage around $1 - \alpha$ and in comparison, when $r$ is greater than 40, `als` fails to guarantee the coverage and `AvgCov` decreases to 0.75.

## 5   Discussion

In this paper, we introduce the conformalized matrix completion method `cmc`, which offers a finite-sample coverage guarantee for all missing entries on average and requires no assumptions on the underlying matrix or any specification of the algorithm adopted, relying only on an estimate of the entrywise sampling probabilities. Given an estimate of the entrywise sampling probability, we provide an upper bound for the coverage gap and give the explicit form with examples of the logistic as well as the general one-bit low-rank missingness. In the implementation, an efficient one-shot weighted conformal approach is proposed with the provable guarantee and achieves nearly exact coverage.

We can compare our findings to a few related results in the literature. The work of Chernozhukov et al. (2021) applies conformal prediction to counterfactual and synthetic control methods, where they include matrix completion as an example with a regression-type formulation. However, their approach relies on a test statistic that is a function of estimated residuals. Consequently, their method requires the assumption of stationarity and weak dependence of errors. Furthermore, the validity of their approach is contingent upon the estimator being either consistent or stable. Additionally, Wieczorek (2023) studies conformal prediction with samples from a deterministic and finite population, but the validity under the sampling without replacement remains an open question in their work.

Our results suggest a number of questions for further inquiry. In the setting of matrix completion, while the results are assumption-free with regard to the matrix $\mathbf{M}$ itself, estimating the sampling probabilities $\mathbf{P}$ that determine which entries are observed remains a key step in the procedure; while our empirical results suggest the method is robust to estimation error in this step, studying the robustness properties more formally is an important open question. More broadly, our algorithm provides an example of how the conformal prediction framework can be applied in a setting that is very different from the usual i.i.d. sampling regime, and may lend insights for how to develop conformal methodologies in other applications with non-i.i.d. sampling structure.

## Acknowledgement

R.F.B. was partially supported by the Office of Naval Research via grant N00014-20-1-2337, and by the National Science Foundation via grant DMS-2023109. C.M. was partially supported by the National Science Foundation via grant DMS-2311127.

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

# Contents

## A   Proofs

The section collects the proofs for the technical results in the main text. Section A.1 and A.2 are devoted to the proof of Lemma 3.1 and Theorem 3.2, respectively. In the end, Section A.3 provides proofs for the coverage gap bounds in the two examples in Section 3.3.

### A.1   Proof of Lemma 3.1

Fix a set of locations $\mathcal{B} := \{(i_1, j_1), \ldots, (i_{n_{\mathrm{cal}}+1}, j_{n_{\mathrm{cal}}+1})\}$, and an index $1 \le m \le n_{\mathrm{cal}} + 1$. By the definition of conditional probability, we have

$$
\mathbb{P}\left((i_*, j_*) = (i_m, j_m) \mid \mathcal{S}_{\mathrm{cal}} \cup \{(i_*, j_*)\} = \mathcal{B}, \ \mathcal{S}_{\mathrm{tr}} = \mathcal{S}_0\right)
$$
$$
= \frac{\mathbb{P}\left((i_*, j_*) = (i_m, j_m), \mathcal{S}_{\mathrm{cal}} = \mathcal{B}\backslash\{(i_m, j_m)\} \mid \mathcal{S}_{\mathrm{tr}} = \mathcal{S}_0, \ |\mathcal{S}| = n\right)}{\mathbb{P}\left(\mathcal{S}_{\mathrm{cal}} \cup \{(i_*, j_*)\} = \mathcal{B} \mid \mathcal{S}_{\mathrm{tr}} = \mathcal{S}_0, \ |\mathcal{S}| = n\right)}
$$
$$
= \frac{\mathbb{P}\left((i_*, j_*) = (i_m, j_m), \mathcal{S}_{\mathrm{cal}} = \mathcal{B}\backslash\{(i_m, j_m)\} \mid \mathcal{S}_{\mathrm{tr}} = \mathcal{S}_0, \ |\mathcal{S}| = n\right)}{\sum_{l=1}^{n_{\mathrm{cal}}+1} \mathbb{P}\left((i_*, j_*) = (i_l, j_l), \mathcal{S}_{\mathrm{cal}} = \mathcal{B}\backslash\{(i_l, j_l)\} \mid \mathcal{S}_{\mathrm{tr}} = \mathcal{S}_0, \ |\mathcal{S}| = n\right)}.
$$

It then boils down to computing $\mathbb{P}\left(\mathcal{S}_{\mathrm{cal}} = \mathcal{S}_1, (i_*, j_*) = (i_l, j_l) \mid \mathcal{S}_{\mathrm{tr}} = \mathcal{S}_0, |\mathcal{S}| = n\right)$ for any fixed set $\mathcal{S}_1$ and fixed location $(i_l, j_l) \notin \mathcal{S}_0 \cup \mathcal{S}_1$. To this end, we have the following claim (we defer its

proof to the end of this section)

$$\mathbb{P}\left(\mathcal{S}_{\mathrm{cal}} = \mathcal{S}_1, (i_*, j_*) = (i_l, j_l) \mid \mathcal{S}_{\mathrm{tr}} = \mathcal{S}_0, |\mathcal{S}| = n\right)$$

$$= \frac{1}{d_1 d_2 - n} \cdot \frac{\prod_{(i,j) \in \mathcal{S}_1} \frac{p_{ij}}{1-p_{ij}}}{\sum_{\mathcal{A} \in \Omega_{\mathcal{S}_{\mathrm{tr}},n}} \prod_{(i',j') \in \mathcal{A}} \frac{p_{i'j'}}{1-p_{i'j'}}}, \tag{8}$$

where $\Omega_{\mathcal{S}_{\mathrm{tr}},n} = \{\mathcal{A} \subseteq [d_1] \times [d_2] : |\mathcal{A}| = n - |\mathcal{S}_{\mathrm{tr}}|, \ \mathcal{A} \cap \mathcal{S}_{\mathrm{tr}} = \varnothing\}$. As a result, we obtain

$$\mathbb{P}\left((i_*, j_*) = (i_m, j_m) \mid \mathcal{S}_{\mathrm{cal}} \cup \{(i_*, j_*)\} = \mathcal{B}, \ \mathcal{S}_{\mathrm{tr}} = \mathcal{S}_0\right)$$

$$= \frac{\prod_{(i,j) \in \mathcal{B} \setminus \{(i_m, j_m)\}} \frac{p_{ij}}{1-p_{ij}}}{\sum_{l=1}^{n_{\mathrm{cal}}+1} \prod_{(i,j) \in \mathcal{B} \setminus \{(i_l, j_l)\}} \frac{p_{ij}}{1-p_{ij}}}$$

$$= \frac{h_{i_m, j_m}}{\sum_{l=1}^{n_{\mathrm{cal}}+1} h_{i_l, j_l}}, \tag{9}$$

where $h_{ij} = (1 - p_{ij})/p_{ij}$. This finishes the proof.

**Proof of Equation** (8). Denote $\mathbf{W}$ the matrix consisting of entries $W_{ij}$ defined in Algorithm 1. The matrix $\mathbf{Z}$ consists of $Z_{ij}$ which are the indicators of non-missingness, i.e. $Z_{ij} = \mathbb{1}\{(i, j) \text{ is observed}\}$. Fix any two disjoint subsets $\mathcal{S}_0, \mathcal{S}_1 \subseteq [d_1] \times [d_2]$ with $|\mathcal{S}_0| + |\mathcal{S}_1| = n$. We have

$$\mathbb{P}\left(\mathcal{S}_{\mathrm{cal}} = \mathcal{S}_1, \mathcal{S}_{\mathrm{tr}} = \mathcal{S}_0 \mid |\mathcal{S}| = n\right)$$
$$= \mathbb{P}\left(\mathrm{supp}(\mathbf{Z}) = \mathcal{S}_0 \cup \mathcal{S}_1, \mathcal{S}_0 \subseteq \mathrm{supp}(\mathbf{W}), \mathcal{S}_1 \subseteq \mathrm{supp}(\mathbf{W})^c \mid |\mathcal{S}| = n\right).$$

Since $\mathbf{W}$ and $\mathbf{Z}$ are independent, one further has

$$\mathbb{P}\left(\mathrm{supp}(\mathbf{Z}) = \mathcal{S}_0 \cup \mathcal{S}_1, \mathcal{S}_0 \subseteq \mathrm{supp}(\mathbf{W}), \mathcal{S}_1 \subseteq \mathrm{supp}(\mathbf{W})^c \mid |\mathcal{S}| = n\right)$$
$$= \mathbb{P}\left(\mathcal{S}_0 \subseteq \mathrm{supp}(\mathbf{W}), \mathcal{S}_1 \subseteq \mathrm{supp}(\mathbf{W})^c\right) \cdot \mathbb{P}\left(\mathrm{supp}(\mathbf{Z}) = \mathcal{S}_0 \cup \mathcal{S}_1 \mid |\mathcal{S}| = n\right)$$
$$= q^{|\mathcal{S}_0|}(1 - q)^{|\mathcal{S}_1|} \cdot \mathbb{P}\left(\mathrm{supp}(\mathbf{Z}) = \mathcal{S}_0 \cup \mathcal{S}_1 \mid |\mathcal{S}| = n\right)$$
$$= \frac{q^{|\mathcal{S}_0|}(1 - q)^{|\mathcal{S}_1|}}{\mathbb{P}(|\mathcal{S}| = n)} \prod_{(i',j') \in [d_1] \times [d_2]} (1 - p_{i'j'}) \prod_{(i,j) \in \mathcal{S}_0 \cup \mathcal{S}_1} \frac{p_{ij}}{1 - p_{ij}}, \tag{10}$$

where the last two identities are based on direct computations.

Based on (10), one can further compute

$$\mathbb{P}\left(\mathcal{S}_{\mathrm{tr}} = \mathcal{S}_0 \mid |\mathcal{S}| = n\right)$$
$$= \sum_{\mathcal{A} \in \Omega_{\mathcal{S}_{\mathrm{tr}},n}} \mathbb{P}\left(\mathcal{S}_{\mathrm{cal}} = \mathcal{A}, \mathcal{S}_{\mathrm{tr}} = \mathcal{S}_0 \mid |\mathcal{S}| = n\right)$$
$$= \frac{q^{|\mathcal{S}_0|}(1 - q)^{n-|\mathcal{S}_0|}}{\mathbb{P}(|\mathcal{S}| = n)} \sum_{\mathcal{A} \in \Omega_{\mathcal{S}_{\mathrm{tr}},n}} \prod_{(i',j') \in [d_1] \times [d_2]} (1 - p_{i'j'}) \prod_{(i,j) \in \mathcal{S}_0 \cup \mathcal{A}} \frac{p_{ij}}{1 - p_{ij}}, \tag{11}$$

where the last identity uses Equation (10).

Now we are ready to prove (8). Recall that the new data point $(i_*, j_*) \mid \mathcal{S}$ is drawn uniformly from $\mathrm{Unif}(\mathcal{S}^c)$. Therefore one has

$$\mathbb{P}\left(\mathcal{S}_{\mathrm{cal}} = \mathcal{S}_1, (i_*, j_*) = (i_{n+1}, j_{n+1}) \mid \mathcal{S}_{\mathrm{tr}} = \mathcal{S}_0, |\mathcal{S}| = n\right)$$
$$= \frac{\mathbb{P}\left(\mathcal{S}_{\mathrm{cal}} = \mathcal{S}_1, \mathcal{S}_{\mathrm{tr}} = \mathcal{S}_0, (i_*, j_*) = (i_{n+1}, j_{n+1}) \mid |\mathcal{S}| = n\right)}{\mathbb{P}\left(\mathcal{S}_{\mathrm{tr}} = \mathcal{S}_0 \mid |\mathcal{S}| = n\right)}$$
$$= \frac{\mathbb{P}\left((i_*, j_*) = (i_{n+1}, j_{n+1}) \mid \mathcal{S} = \mathcal{S}_1 \cup \mathcal{S}_0\right) \mathbb{P}\left(\mathcal{S}_{\mathrm{cal}} = \mathcal{S}_1, \mathcal{S}_{\mathrm{tr}} = \mathcal{S}_0 \mid |\mathcal{S}| = n\right)}{\mathbb{P}\left(\mathcal{S}_{\mathrm{tr}} = \mathcal{S}_0 \mid |\mathcal{S}| = n\right)}$$
$$= \frac{1}{d_1 d_2 - n} \cdot \frac{\prod_{(i,j) \in \mathcal{S}_1} \frac{p_{ij}}{1-p_{ij}}}{\sum_{\mathcal{A} \in \Omega_{\mathcal{S}_{\mathrm{tr}},n}} \prod_{(i',j') \in \mathcal{A}} \frac{p_{i'j'}}{1-p_{i'j'}}}, \tag{12}$$

where the last line follows from (10) and (11).

## A.2 Proof of Theorem 3.2

First, fix any $a \in [0, 1]$. Lemma 3.1, together with the weighted conformal prediction framework of Tibshirani et al. (2019), implies that

$$\mathbb{P}\left(M_{i_* j_*} \in \widehat{M}_{i_* j_*} \pm q^*_{i_* j_*}(a) \cdot \widehat{s}_{i_* j_*} \mid \mathcal{S}_{\mathrm{tr}}, \mathcal{S}_{\mathrm{cal}} \cup \{(i_*, j_*)\}\right) \geq 1 - a,$$

where

$$q^*_{i_* j_*}(a) = \mathrm{Quantile}_{1-a}\left(\sum_{(i,j) \in \mathcal{S}_{\mathrm{cal}} \cup \{(i_*, j_*)\}} w_{ij} \cdot \delta_{R_{ij}}\right),$$

and

$$w_{ij} = \frac{h_{ij}}{\sum_{(i',j') \in \mathcal{S}_{\mathrm{cal}} \cup \{(i_*, j_*)\}} h_{i'j'}}.$$

Indeed, here $a$ can be any function of the random variables we are conditioning on—that is, $a$ may depend on $n$, on $\mathcal{S}_{\mathrm{tr}}$, and on $\mathcal{S}_{\mathrm{cal}} \cup \{(i_*, j_*)\}$.

Next define $a = \alpha + \Delta$ where $\Delta$ is defined as in (6). We observe that, since each $\widehat{h}_{ij}$ is a function of $\mathcal{S}_{\mathrm{tr}}$, then $\Delta$ (and thus also $a$) can therefore be expressed as a function of $n$, $\mathcal{S}_{\mathrm{tr}}$, and $\mathcal{S}_{\mathrm{cal}} \cup \{(i_*, j_*)\}$. Therefore, applying the work above, we have

$$\mathbb{P}\left(M_{i_* j_*} \in \widehat{M}_{i_* j_*} \pm q^*_{i_* j_*}(\alpha + \Delta) \cdot \widehat{s}_{i_* j_*} \mid \mathcal{S}_{\mathrm{tr}}, \mathcal{S}_{\mathrm{cal}} \cup \{(i_*, j_*)\}\right) \geq 1 - \alpha - \Delta$$

and thus, after marginalizing,

$$\mathbb{P}\left(M_{i_* j_*} \in \widehat{M}_{i_* j_*} \pm q^*_{i_* j_*}(\alpha + \Delta) \cdot \widehat{s}_{i_* j_*}\right) \geq 1 - \alpha - \mathbb{E}[\Delta].$$

Next, we verify that

$$\widehat{q} \geq q^*_{i_* j_*}(\alpha + \Delta)$$

holds almost surely—if this is indeed the case, then we have shown that

$$\mathbb{P}\left(M_{i_* j_*} \in \widehat{M}_{i_* j_*} \pm \widehat{q} \cdot \widehat{s}_{i_* j_*}\right) \geq 1 - \alpha - \mathbb{E}[\Delta],$$

which establishes the desired result. Thus we only need to show that $\widehat{q} \geq q^*_{i_* j_*}(\alpha + \Delta)$, or equivalently,

$$\mathrm{Quantile}_{1-\alpha}\left(\sum_{(i,j) \in \mathcal{S}_{\mathrm{cal}}} \widehat{w}_{ij} \cdot \delta_{R_{ij}} + \widehat{w}_{\mathrm{test}} \cdot \delta_{+\infty}\right) \geq \mathrm{Quantile}_{1-\alpha-\Delta}\left(\sum_{(i,j) \in \mathcal{S}_{\mathrm{cal}} \cup \{(i_*, j_*)\}} w_{ij} \cdot \delta_{R_{ij}}\right).$$

Define

$$w'_{ij} = \frac{\widehat{h}_{ij}}{\sum_{(i',j') \in \mathcal{S}_{\mathrm{cal}} \cup \{(i_*, j_*)\}} \widehat{h}_{i'j'}} \tag{13}$$

for all $(i, j) \in \mathcal{S} \cup \{(i_*, j_*)\}$. Then by definition of $\widehat{w}$, we see that $w'_{ij} \geq \widehat{w}_{ij}$ for $(i, j) \in \mathcal{S}$, and therefore,

$$\mathrm{Quantile}_{1-\alpha}\left(\sum_{(i,j) \in \mathcal{S}_{\mathrm{cal}}} \widehat{w}_{ij} \cdot \delta_{R_{ij}} + \widehat{w}_{\mathrm{test}} \cdot \delta_{+\infty}\right)$$

$$\geq \mathrm{Quantile}_{1-\alpha}\left(\sum_{(i,j) \in \mathcal{S}_{\mathrm{cal}}} w'_{ij} \cdot \delta_{R_{ij}} + w'_{(i_*, j_*)} \cdot \delta_{+\infty}\right) \geq \mathrm{Quantile}_{1-\alpha}\left(\sum_{(i,j) \in \mathcal{S}_{\mathrm{cal}} \cup \{(i_*, j_*)\}} w'_{ij} \cdot \delta_{R_{ij}}\right)$$

holds almost surely. Therefore it suffices to show that

$$\mathrm{Quantile}_{1-\alpha}\left(\sum_{(i,j) \in \mathcal{S}_{\mathrm{cal}} \cup \{(i_*, j_*)\}} w'_{ij} \cdot \delta_{R_{ij}}\right) \geq \mathrm{Quantile}_{1-\alpha-\Delta}\left(\sum_{(i,j) \in \mathcal{S}_{\mathrm{cal}} \cup \{(i_*, j_*)\}} w_{ij} \cdot \delta_{R_{ij}}\right)$$

holds almost surely. Indeed, we have

$$\mathsf{d}_{\mathsf{TV}}\left(\sum_{(i,j) \in \mathcal{S}_{\mathrm{cal}} \cup \{(i_*, j_*)\}} w'_{ij} \cdot \delta_{R_{ij}}, \sum_{(i,j) \in \mathcal{S}_{\mathrm{cal}} \cup \{(i_*, j_*)\}} w_{ij} \cdot \delta_{R_{ij}}\right) \leq \frac{1}{2} \sum_{(i,j) \in \mathcal{S}_{\mathrm{cal}} \cup \{(i_*, j_*)\}} |w'_{ij} - w_{ij}| = \Delta,$$

where $\mathsf{d}_{\mathsf{TV}}$ denotes the total variation distance. This completes the proof.

## A.3 Proofs for examples in Section 3.3

Recall the definition of $\Delta$:

$$\Delta = \frac{1}{2} \sum_{(i,j) \in \mathcal{S}_{\mathrm{cal}} \cup \{(i_*,j_*)\}} \left| \frac{\widehat{h}_{ij}}{\sum_{(i',j') \in \mathcal{S}_{\mathrm{cal}} \cup \{(i_*,j_*)\}} \widehat{h}_{i'j'}} - \frac{h_{ij}}{\sum_{(i',j') \in \mathcal{S}_{\mathrm{cal}} \cup \{(i_*,j_*)\}} h_{i'j'}} \right|.$$

Note that $\Delta \leq 1$ by definition. We start with stating a useful lemma to bound the coverage gap $\Delta$ using the estimation error of $\widehat{h}_{ij}$.

**Lemma A.1.** *By the definition of the coverage gap $\Delta$, we have*

$$\Delta \leq \frac{\sum_{(i,j) \in \mathcal{S}_{\mathrm{cal}} \cup \{(i_*,j_*)\}} |\widehat{h}_{ij} - h_{ij}|}{\sum_{(i',j') \in \mathcal{S}_{\mathrm{cal}}} \widehat{h}_{i'j'}}. \tag{14}$$

### A.3.1 Proof for Example 3.3.1

Under the logistic model, one has for any $(i,j)$

$$\mathbb{P}((i,j) \in \mathcal{S}_{\mathrm{tr}}) = q \cdot p_{ij} = \frac{q \cdot \exp(u_i + v_j)}{1 + \exp(u_i + v_j)}.$$

In this case, if $\widehat{u}$ and $\widehat{v}$ are the constrained maximum likelihood estimators in Example 3.3.1, Theorem 6 in Chen et al. (2023) implies that

$$\|\widehat{\mathbf{u}} - \mathbf{u}\|_\infty = O_\mathbb{P}\left(\sqrt{\frac{\log d_1}{d_2}}\right), \qquad \|\widehat{\mathbf{v}} - \mathbf{v}\|_\infty = O_\mathbb{P}\left(\sqrt{\frac{\log d_2}{d_1}}\right),$$

with the proviso that $\|u\|_\infty + \|v\|_\infty \leq \tau < \infty$, $d_2 \gg \sqrt{d_1} \log d_1$ and $d_1 \gg (\log d_2)^2$. Then for $h_{ij} = \exp(-u_i - v_j)$ and $\widehat{h}_{ij} = \exp(-\widehat{u}_i - \widehat{v}_j)$, we have

$$\max_{i,j} |\widehat{h}_{ij} - h_{ij}| = \max_{i,j} e^{-u_i - v_j} \left( e^{-(\widehat{u}_i - u_i) - (\widehat{v}_j - v_j)} - 1 \right) = O_\mathbb{P}\left(\sqrt{\frac{\log d_1}{d_2}} + \sqrt{\frac{\log d_2}{d_1}}\right).$$

Further, as $\min_{i,j} h_{i,j} = \min_{i,j} \exp(-u_i - v_j) \geq e^{-\tau}$, then with probability approaching one, for every $(i,j) \in [d_1] \times [d_2]$, one has $\widehat{h}_{i,j} \geq h_{i,j} - |h_{ij} - \widehat{h}_{ij}| \geq e^{-\tau}/2 =: h_0$. By the upper bound (14), we have

$$\Delta \lesssim \sqrt{\frac{\log d_1}{d_2}} + \sqrt{\frac{\log d_2}{d_1}} \asymp \sqrt{\frac{\log \max\{d_1, d_2\}}{\min\{d_1, d_2\}}}.$$

Further, as $\Delta \leq 1$, we have $\mathbb{E}[\Delta] \lesssim \sqrt{\frac{\log \max\{d_1,d_2\}}{\min\{d_1,d_2\}}}$.

### A.3.2 Proof of Example 3.3.2

Define the link function $\psi(t) = q(1 + e^{-\phi(t)})$, where $\phi$ is monotonic. Applying Theorem 1 in the paper (Davenport et al., 2014), we obtain that with probability at least $1 - C_1/(d_1 + d_2)$

$$\frac{1}{d_1 d_2} \|\widehat{\mathbf{A}} - \mathbf{A}\|_\mathrm{F}^2 \leq \sqrt{2}\widetilde{C}_\tau \sqrt{\frac{k(d_1 + d_2)}{d_1 d_2}}, \tag{15}$$

with the proviso that $d_1 d_2 \geq (d_1 + d_2) \log(d_1 d_2)$. Here $\widetilde{C}_\tau = 2^{39/4} e^{9/4} (1 + \sqrt{6}) \tau L_\tau \beta_\tau$ with

$$L_\tau = \sup_{-\tau \leq t \leq \tau} \frac{|\psi'(t)|}{\psi(t)(1 - \psi(t))}, \quad \text{and} \quad \beta_\tau = \sup_{-\tau \leq t \leq \tau} \frac{\psi(t)(1 - \psi(t))}{|\psi'(t)|^2}. \tag{16}$$

Denote this high probability event to be $\mathcal{E}_0$. Since $\|\mathbf{A}\|_1 \leq \sqrt{d_1 d_2}\|\mathbf{A}\|_\mathrm{F}$, on this event $\mathcal{E}_0$, we further have

$$\frac{1}{d_1 d_2} \|\widehat{\mathbf{A}} - \mathbf{A}\|_1 \leq C_\tau \left(\frac{k(d_1 + d_2)}{d_1 d_2}\right)^{1/4},$$

where $C_\tau = \zeta\sqrt{L_\tau\beta_\tau}$ and $\zeta$ is a universal constant.

Recall that $h_{ij} = \exp(-\phi(A_{ij}))$ and $\|\mathbf{A}\|_\infty \leq \tau$. On the same event $\mathcal{E}_0$, we further have

$$\frac{1}{d_1 d_2}\|\widehat{\mathbf{H}} - \mathbf{H}\|_1 \leq C'_\tau \left(\frac{k(d_1 + d_2)}{d_1 d_2}\right)^{1/4}, \tag{17}$$

where $C'_\tau = 2e^{\phi(\tau)}\sqrt{\widetilde{C}_\tau}$.

By the feasibility of the minimizer $\widehat{\mathbf{A}}$ and the fact that $\widehat{h}_{ij} = \exp(-\phi(\widehat{A}_{ij}))$, we have $\widehat{h}_{ij} \geq h_0 = e^{-\phi(\tau)}$ for all $(i, j)$. This together with the upper bound (14) implies that

$$\Delta \leq \frac{1}{h_0 n_{\mathrm{cal}}}\sum_{(i,j)\in\mathcal{S}_{\mathrm{cal}}\cup\{(i_*,j_*)\}}|h_{ij} - \widehat{h}_{ij}| \leq \frac{1}{h_0 n_{\mathrm{cal}}}\|\widehat{\mathbf{H}} - \mathbf{H}\|_1.$$

Define a second high probability event $\mathcal{E}_1 = \{n_{\mathrm{cal}} \geq (1-c)(1-q)\|\mathbf{P}\|_1\}$. Using the Chernoff bound, we have $\mathbb{P}(\mathcal{E}_1) \geq 1 - C_0(d_1 d_2)^{-5}$. Therefore, on the event $\mathcal{E}_0 \cap \mathcal{E}_1$, we have

$$\Delta \leq \frac{d_1 d_2}{ch_0(1-q)\|\mathbf{P}\|_1}C'_\tau\left(\frac{k(d_1+d_2)}{d_1 d_2}\right)^{1/4}.$$

Under the assumptions that $p_{ij} = 1/(1 + e^{-\phi(A_{ij})})$, and that $\|\mathbf{A}\|_\infty \leq \tau$, we know that $p_{ij} \geq C_2$ for some constant that only depends on $\tau$. As a result, we have $\|\mathbf{P}\|_1 \geq C_2 d_1 d_2$, which further leads to the conclusion that

$$\Delta \leq \frac{1}{cC_2 h_0(1-q)}C'_\tau\left(\frac{k(d_1+d_2)}{d_1 d_2}\right)^{1/4}.$$

on the event $\mathcal{E}_0 \cap \mathcal{E}_1$. In addition, on the small probability event $(\mathcal{E}_0 \cap \mathcal{E}_1)$, one trivially has $\Delta \leq 1$. Therefore simple combinations of the cases yields the desired bound on $\mathbb{E}[\Delta]$.

### A.3.3   Proof of Lemma A.1

Reusing the definition of $w'$ (13), one has

$$\begin{aligned}
\Delta &= \frac{1}{2}\sum_{(i,j)\in\mathcal{S}_{\mathrm{cal}}\cup\{(i_*,j_*)\}}|w_{ij} - w'_{ij}| \\
&= \frac{1}{2}\sum_{(i,j)\in\mathcal{S}_{\mathrm{cal}}\cup\{(i_*,j_*)\}}\frac{\left|h_{ij}\sum_{(i',j')\in\mathcal{S}_{\mathrm{cal}}\cup\{(i_*,j_*)\}}\widehat{h}_{i'j'} - \widehat{h}_{ij}\sum_{(i',j')\in\mathcal{S}_{\mathrm{cal}}\cup\{(i_*,j_*)\}}h_{i'j'}\right|}{\left(\sum_{(i',j')\in\mathcal{S}_{\mathrm{cal}}\cup\{(i_*,j_*)\}}\widehat{h}_{i'j'}\right)\left(\sum_{(i',j')\in\mathcal{S}_{\mathrm{cal}}\cup\{(i_*,j_*)\}}h_{i'j'}\right)} \\
&\leq \frac{1}{2}\sum_{(i,j)\in\mathcal{S}_{\mathrm{cal}}\cup\{(i_*,j_*)\}}\left\{\frac{h_{ij}\left|\sum_{(i',j')\in\mathcal{S}_{\mathrm{cal}}\cup\{(i_*,j_*)\}}\widehat{h}_{i'j'} - \sum_{(i',j')\in\mathcal{S}_{\mathrm{cal}}\cup\{(i_*,j_*)\}}h_{i'j'}\right|}{\left(\sum_{(i',j')\in\mathcal{S}_{\mathrm{cal}}\cup\{(i_*,j_*)\}}\widehat{h}_{i'j'}\right)\left(\sum_{(i',j')\in\mathcal{S}_{\mathrm{cal}}\cup\{(i_*,j_*)\}}h_{i'j'}\right)}\right. \\
&\qquad\left. + \frac{|\widehat{h}_{ij} - h_{ij}|\sum_{(i',j')\in\mathcal{S}_{\mathrm{cal}}\cup\{(i_*,j_*)\}}h_{i'j'}}{\left(\sum_{(i',j')\in\mathcal{S}_{\mathrm{cal}}\cup\{(i_*,j_*)\}}\widehat{h}_{i'j'}\right)\left(\sum_{(i',j')\in\mathcal{S}_{\mathrm{cal}}\cup\{(i_*,j_*)\}}h_{i'j'}\right)}\right\} \\
&= \frac{1}{2}\frac{\left|\sum_{(i',j')\in\mathcal{S}_{\mathrm{cal}}\cup\{(i_*,j_*)\}}\widehat{h}_{i'j'} - \sum_{(i',j')\in\mathcal{S}_{\mathrm{cal}}\cup\{(i_*,j_*)\}}h_{i'j'}\right|}{\left(\sum_{(i',j')\in\mathcal{S}_{\mathrm{cal}}\cup\{(i_*,j_*)\}}\widehat{h}_{i'j'}\right)} + \frac{1}{2}\frac{\sum_{(i,j)\in\mathcal{S}_{\mathrm{cal}}\cup\{(i_*,j_*)\}}|\widehat{h}_{ij} - h_{ij}|}{\left(\sum_{(i',j')\in\mathcal{S}_{\mathrm{cal}}\cup\{(i_*,j_*)\}}\widehat{h}_{i'j'}\right)} \\
&\leq \frac{\sum_{(i,j)\in\mathcal{S}_{\mathrm{cal}}\cup\{(i_*,j_*)\}}|\widehat{h}_{ij} - h_{ij}|}{\sum_{(i',j')\in\mathcal{S}_{\mathrm{cal}}\cup\{(i_*,j_*)\}}\widehat{h}_{i'j'}}. \tag{18}
\end{aligned}$$

This completes the proof.

## B   Additional numerical experiments

In this section, we provide additional simulation results. In Section B.1, we compare the proposed one-shot weighted approach with the exact weighted conformal prediction. In Section B.2, the

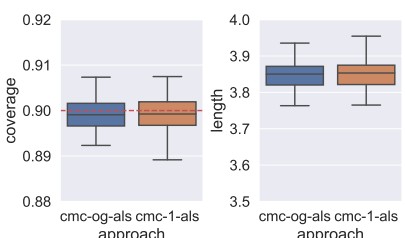 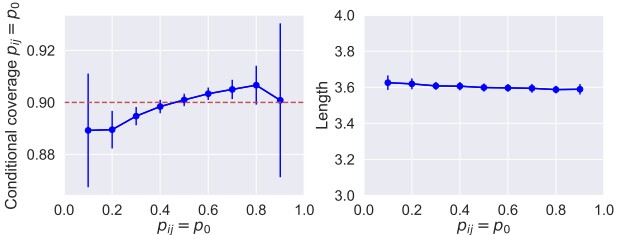

(a) Exact/original (og) `cmc` vs 1-shot `cmc`.  (b) Local performance of coverage given $p_{ij} = p_0$.

local coverage of `cmc-als` conditioning on $p_{ij}$ is evaluated. We present the simulation results for the convex relaxation (`cvx`) and the conformalized convex method (`cmc-cvx`) in both settings with homogeneous and heterogeneous missingness in Section B.3 and B.5. We further evaluate the performance of `cmc-als` when the sampling probability is misspecified in Section B.6.

## B.1  Comparison between one-shot and exact weighted approaches

We conduct comparison in the setting with heterogeneous missingness and Gaussian noise as shown in Section 4.1.2 with $k^* = 12$. From the result in Figure 5a, the performance of two aproaches are essentially the same while the one-shot `cmc-als` is more computationally efficient.

## B.2  Local coverage of `cmc`

In Figure 5b, we evaluate the local performance of `cmc-als` when conditioning on $\{p_{ij} = p_0\}$, i.e. a subpopulation determined by the value of sampling probability. We use the setting with heterogeneous missingness and Gaussian noise as shown in Section 4.1.2 with $k^* = 12$ and $p_0$ ranging from $0.1$ to $0.9$ with a step size of $0.1$. The conditional coverage is approximated via kernel smoothing: with the indicators $A_{ij}$ for coverage, we use the weight $K_{ij} = \phi_{p_0,h}(p_{ij})$ and calculate the conditional coverage by $(\sum A_{ij} K_{ij})/(\sum K_{ij})$. Here $\phi_{\mu,\sigma}$ is the density function of $\mathcal{N}(\mu, \sigma^2)$ and $h = 0.05$. When $p_0$ increases from $0.2$ to $0.8$, the conditional coverage increases and stays around $0.9$. At $p_0 = 0.1, 0.9$, the coverage varies much, which is due to the small effective sample size for the edge values of $p_{ij}$. Moreover, for uniform sampling, since the rows and columns are generated i.i.d., there are no meaningful subgroups of the data to condition on.

## B.3  Additional results for homogeneous misingness

In this section, we present the results for synthetic simulation with the convex relaxation `cvx` and the conformalized convex matrix completion method `cmc-cvx`. Setting 1, 2, 3, 4 are the same as introduced in Section4.1. The true rank is $r^* = 8$ and the hypothesized rank varies from $4$ to $24$ with the stepsize $4$.

The conformalized methods, regardless of the based algorithm adopted, have nearly exact coverage around $1 - \alpha$. But we can observe different behaviors between `als` and `cvx` since the convex relaxation is free of the choice of $r$ until the projection of $\widehat{\mathbf{M}}_{\mathrm{cvx}}$ onto the rank-$r$ subspace (Chen et al., 2020). As a result, when $r > r^*$, `cvx` tends to overestimate the strength of the noise. In Figure 6a, 6b and 6d, when $r > r^*$, `cvx` has coverage rate higher than the target level and the confidence interval is more conservative than conformalized methods. Since the accuracy of `cvx` is based on the large sample size, in Figure 6b, when the effective sample size is insufficient with small $p_{ij}$, the residuals from `cvx` have a large deviation from the standard distribution and the intervals are much larger than the oracle ones. Besides, in Figure 6c, when the noise has heavy tails than Gaussian variables, `cvx` overestimates the noise strength similar to `als` and is conservative in coverage. When the incoherence condition is violated in Figure 6d, if $r < r^*$, both `cvx` and `als` fit the missed factor by overestimating the noise strength and produce extremely large intervals.

## B.4  Estimation error for one-bit matrix estimation

The estimation error in $\widehat{p}_{ij}$ can be visualized from the following heatmaps comparing $\mathbf{P}$ and $\widehat{\mathbf{P}}$. Here the entries are sorted by the order of $p_{ij}$ for each row and each column.

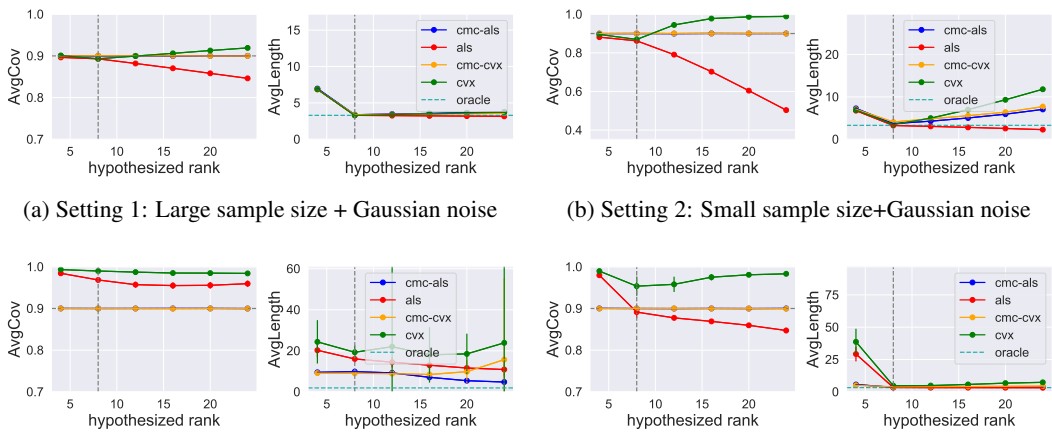

(a) Setting 1: Large sample size + Gaussian noise

(b) Setting 2: Small sample size+Gaussian noise

(c) Setting 3: Large sample size + heavy-tailed noise

(d) Setting 4: Violation of incoherence

Figure 6: Comparison between conformalized and model-based matrix completion approaches.

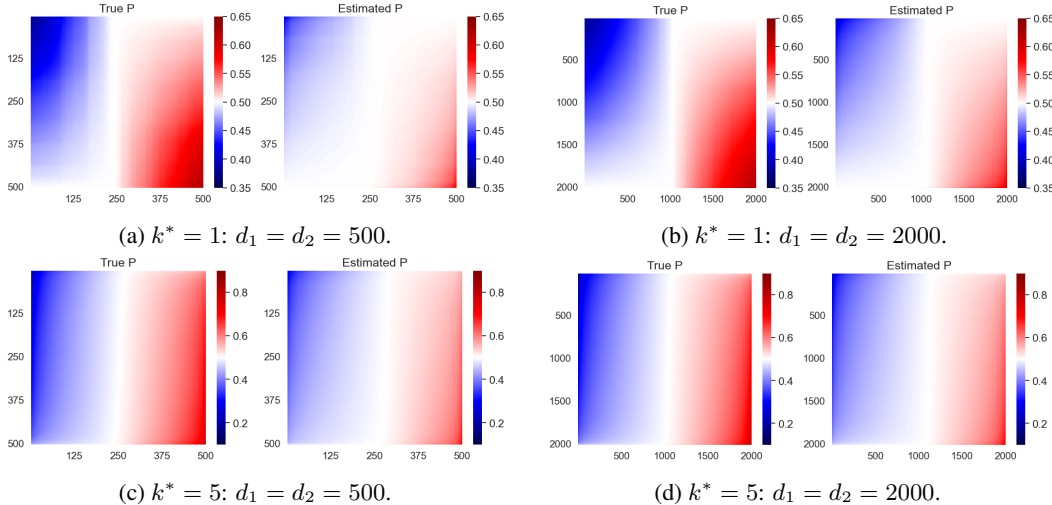

(a) $k^* = 1$: $d_1 = d_2 = 500$.

(b) $k^* = 1$: $d_1 = d_2 = 2000$.

(c) $k^* = 5$: $d_1 = d_2 = 500$.

(d) $k^* = 5$: $d_1 = d_2 = 2000$.

Figure 7: Heatmaps for $\mathbf{P}$ and $\widehat{\mathbf{P}}$.

### B.5 Additional results for heterogeneous missingness

In Figure 8, we present the results for synthetic simulation with the convex relaxation `cvx` as the base algorithm, where we denote `cmc-cvx` and `cmc*-cvx` as the conformalized matrix completion method with estimated weights and true weights, respectively. Three settings with heterogeneous missingness are the same as Figure 3.

### B.6 Misspecified sampling probability

We consider the following four settings:

(a) The underlying missingness follows the rank-one model where $p_{ij} = a_i b_j$, both $a_i$ and $b_j$ are generated i.i.d. from $\text{Unif}(0.2, 1)$. The noise is adversarial. But we estimate $p_{ij}$ via the one-bit matrix completion based on the logistic model (working model with hypothesized rank $k = 5$).

(b) The underlying missingness follows the logistic model as in Example 3.3.2 with $k^* = 5$ and we adopt the adversarial noise. But $p_{ij}$ is estimated under the assumption of uniform sampling (working model), i.e. $\widehat{p}_{ij} = \widehat{p}$.

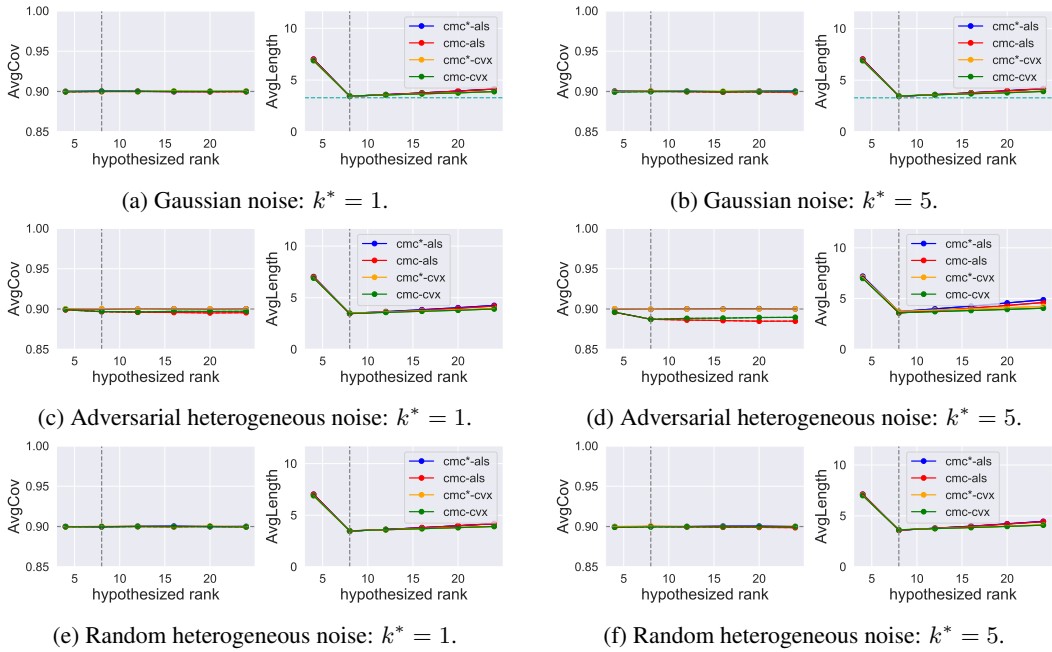

(a) Gaussian noise: $k^* = 1$.

(b) Gaussian noise: $k^* = 5$.

(c) Adversarial heterogeneous noise: $k^* = 1$.

(d) Adversarial heterogeneous noise: $k^* = 5$.

(e) Random heterogeneous noise: $k^* = 1$.

(f) Random heterogeneous noise: $k^* = 5$.

Figure 8: Comparison under heterogenous missingness.

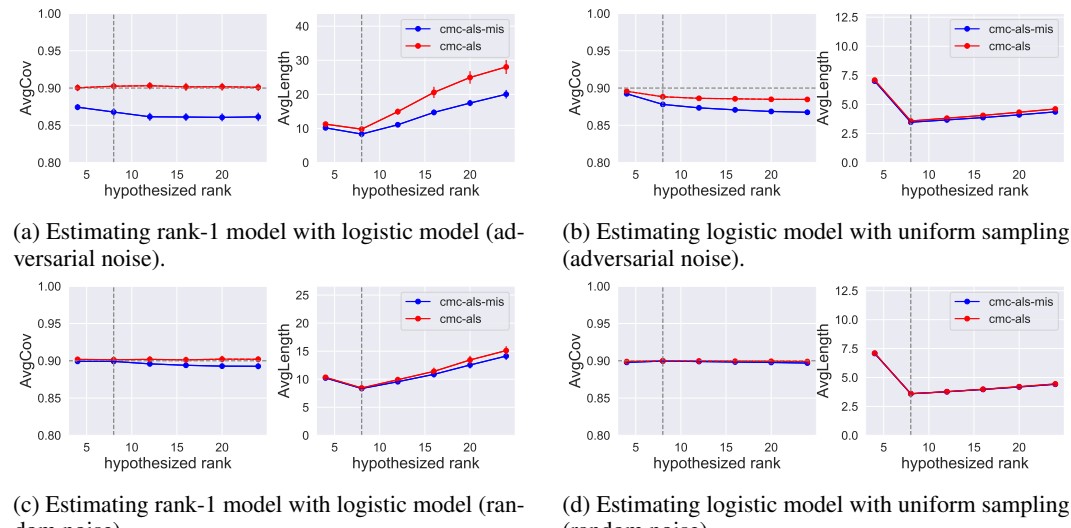

(a) Estimating rank-1 model with logistic model (adversarial noise).

(b) Estimating logistic model with uniform sampling (adversarial noise).

(c) Estimating rank-1 model with logistic model (random noise).

(d) Estimating logistic model with uniform sampling (random noise).

(c) Same as (a) except that we use the random heterogeneous noise.

(d) Same as (b) except that we use the random heterogeneous noise.

From the results, we can see that when the noise is generated following the random heterogeneous model, where the values of entries are independent of the sampling probabilities, the misspecification of the sampling model only slightly affects the coverage. Moreover, when the noise is generated in the adversarial way, where the values of entries depend on $p_{ij}$'s, we can see that the coverage with a misspecified sampling model is lower than the target level, but is above $0.85$ in practice, which depends on the divergence between the true and the working sampling models.

## B.7 Additional results for the sales dataset

Denote $\mathbf{M}$ the underlying matrix in the sales dataset. In Figure 10a, we plot singular values of $\mathbf{M}$ and top-5 singular values contain a large proportion of the information. In Figure 10b, we plot the histogram of entries $M_{ij}$'s of the underlying matrix, and the sales dataset has the range from 0 to over 20 thousand with a heavy tail.

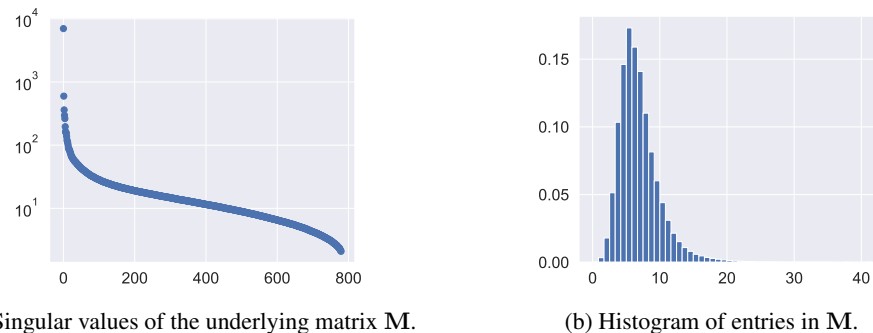

(a) Singular values of the underlying matrix $\mathbf{M}$.      (b) Histogram of entries in $\mathbf{M}$.

Figure 10: Descriptive plots for the underlying matrix $\mathbf{M}$.

### B.7.1 Comparison under misspecified missingness

We consider the sales dataset where the missingness occurs in the way that: for weekdays, each entry is observed with $p = 0.8$. On weekends, as the stores are likely to be operated by less experienced interns or are less frequent to report the sales data, each entry is observed with a lower probability, e.g. $0.8/3$. Moreover, as there could be a subgroup of stores that are less frequent in sales reporting, 200 stores are randomly sampled, for which $p = 0.8/3$. We use the logistic model with $k = 5$ to estimate $p_{ij}$. Comparison between `als` and `cmc-als` is shown in Figure 11.

### B.7.2 Results with convex optimization

In Figure 12, `cmc-cvx` has nearly exact coverage at $1 - \alpha$, but `cvx` tends to have higher coverage than the target level. Besides, the convex approach has much larger intervals when $r$ is large, which can be caused by the overfitting of the observed entries. As conformalized approach leaves out a proportion of observed entries as the training set, intervals produced by `cmc-cvx` are less accurate than `cvx` due to the poorly behaved base algorithm.

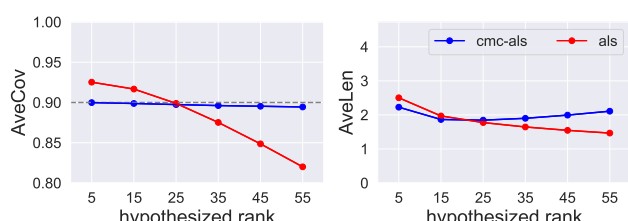

Figure 11: Sales dataset with heterogeneous and misspecified missingness.

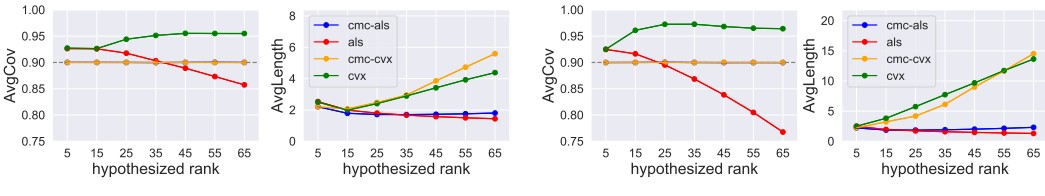

(a) Homogeneous missingness.  (b) Heterogeneous missingness with $k^* = 1$.

Figure 12: Comparison between conformalized and model-based matrix completion with sales dataset.

## C  Additional details of algorithms and extensions

### C.1  Extension to likelihood-based scores and categorical matrices

We will show in this section that cmc can also be applied to categorical matrix completion (Cao and Xie, 2015) or a more general setting in (19), the validity of which is also guaranteed by the presented theorem.

**Setup**  To formulate the problem, consider an underlying parameter matrix $\mathbf{M}^* \in [d_1] \times [d_2]$ and the observations $\{M_{ij} : i \in [d_1], j \in [d_2]\}$ are drawn from the distribution

$$M_{ij} \mid M_{ij}^* \sim \mathcal{P}_{M_{ij}^*}, \tag{19}$$

where $\{\mathcal{P}_\theta\}_{\theta \in \Theta}$ can be a family of parametric distributions with probability density $p_\theta$. The categorical matrix completion is a specific example where the support of $M_{ij}$ is finite or countable. For example, a Poisson matrix is generated by $M_{ij} \sim \text{Pois}(M_{ij}^*)$, where $M_{ij}^*$ is the Poisson mean. Similar to the previous setup, we treat $\mathbf{M}$ as deterministic, and entries in the subset $\mathcal{S} \subseteq [d_1] \times [d_2]$ are available. Here $\mathcal{S}$ is sampled in the same manner as before with the matrix $\mathbf{P} \in [d_1] \times [d_2]$.

**Split conformal approach**  Consider the split approach with the partition $\mathcal{S} = \mathcal{S}_{\text{tr}} \cup \mathcal{S}_{\text{cal}}$ as in Algorithm 1. With the training set $\mathbf{M}_{\text{tr}}$, we obtain an estimated likelihood function $\widehat{\pi}(m; i, j)$ such that

$$\widehat{\pi}(m; i, j) = \widehat{p}_{M_{ij}^*}(m),$$

which is an estimate for the true likelihood of $M_{ij}$ at $m$. The estimation can be feasible given certain low-complexity structures. For example, if a hypothesized distribution family $\{\mathcal{Q}_\theta\}_{\theta \in \Theta}$ with probability density $q_\theta$ is given and the underlying mean matrix $\mathbf{M}^*$ is assumed to be low-rank. Then, $\mathbf{M}$ can be viewed as a perturbation of $\mathbf{M}^*$ and we can estimate $\mathbf{M}^*$ via matrix completion algorithms with entries in $\mathbf{M}_{\text{tr}}$. Denote $\widehat{\mathbf{M}}$ as the estimate for $\mathbf{M}^*$, then we have the estimated likelihood

$$\widehat{\pi}(m; i, j) = q_{\widehat{M}_{ij}}(m).$$

The odds ratios are also estimated from the training set, i.e. $\widehat{h}_{ij}$, from which we compute the weights

$$\widehat{w}_{ij} = \frac{\widehat{h}_{ij}}{\displaystyle\sum_{(i',j') \in \mathcal{S}_{\text{cal}}} \widehat{h}_{i'j'} + \max_{(i',j') \in \mathcal{S}^c} \widehat{h}_{i'j'}}, \ (i,j) \in \mathcal{S}_{\text{cal}}, \quad \widehat{w}_{\text{test}} = \frac{\widehat{h}_{i_* j_*}}{\displaystyle\sum_{(i',j') \in \mathcal{S}_{\text{cal}}} \widehat{h}_{i'j'} + \max_{(i',j') \in \mathcal{S}^c} \widehat{h}_{i'j'}}.$$

For each $(i, j) \in \mathcal{S}_{\text{cal}}$, calculate the likelihood-based nonconformity score

$$R_{ij} = -\widehat{\pi}(M_{ij}; i, j).$$

Then, for any test point $(i_*, j_*) \in \mathcal{S}^c$, we can construct the confidence interval

$$\widehat{C}(i_*, j_*) = \{m \in [K] : \ \widehat{\pi}(m; i_*, j_*) \leq \widehat{q}\},$$

where $\widehat{q}$ is the weighted quantile

$$\widehat{q} = \text{Quantile}_{1-\alpha} \left( \sum_{(i,j) \in \mathcal{S}_{\text{cal}}} \widehat{w}_{ij} \cdot \delta_{R_{ij}} + \widehat{w}_{\text{test}} \cdot \delta_{+\infty} \right).$$

More examples of conformal methods for classification are shown in Romano et al. (2020), Angelopoulos et al. (2020), etc.

## C.2  Full conformalized matrix completion

In Algorithm 2, the procedure of the full conformalized matrix completion (`full-cmc`) is presented. This full conformal version of `cmc` offers the same coverage guarantee as given in Theorem 3.2 for the split version of `cmc` (except with the entire observed set $\mathcal{S}$ in place of $\mathcal{S}_{\mathrm{cal}}$, when defining $\Delta(\widehat{\boldsymbol{w}})$; the formal proof of this bound for full conformal is very similar to the proof of Theorem 3.2, using an analogous weighted exchangeability argument as in Lemma 3.1, and so we omit it here.

To define this algorithm, we need some notation: given the observed data $\mathbf{M}_{\mathcal{S}}$, plus a test point location $(i_*, j_*)$ and a hypothesized value $m$ for the test point $M_{i_* j_*}$, define a matrix $\mathbf{M}^{(m)}$ with entries

$$
M_{ij}^{(m)} = \begin{cases} M_{ij}, & (i,j) \in \mathcal{S}, \\ m, & (i,j) = (i_*, j_*), \\ \emptyset, & \text{otherwise,} \end{cases} \tag{20}
$$

where, abusing notation, "$M_{ij} = \emptyset$" denotes that no information is observed in this entry.

---

**Algorithm 2** `full-cmc`: full conformalized matrix completion

---

1: **Input**: target level $1 - \alpha$; partially observed matrix $\mathbf{M}_{\mathcal{S}}$.
2: Using the training data $\mathbf{M}_{\mathcal{S}}$, compute an estimate $\widehat{\mathbf{P}}$ of the observation probabilities (with $\widehat{p}_{ij}$ estimating $p_{ij}$, the probability of entry $(i,j)$ being observed).
3: **for** $(i_*, j_*)$ in $\mathcal{S}^c$ **do**
4:    **for** $m \in \mathcal{M}$ **do**
5:       Augment $\mathbf{M}_{\mathcal{S}}$ with one additional hypothesized entry, $\{M_{i_*, j_*} = m\}$, to obtain $\mathbf{M}^{(m)}$ defined as in (20).
6:       Using the imputed matrix $\mathbf{M}_{\mathcal{S}}^{(m)}$, compute:

   - An initial estimate $\widehat{\mathbf{M}}^{(m)}$ using any matrix completion algorithm (with $\widehat{M}_{ij}^{(m)}$ estimating the target $M_{ij}$);

   - Optionally, a local uncertainty estimate $\widehat{\mathbf{s}}^{(m)}$ (with $\widehat{s}_{ij}^{(m)}$ estimating our relative uncertainty in the estimate $\widehat{M}_{ij}^{(m)}$), or otherwise set $\widehat{s}_{ij}^{(m)} \equiv 1$;

   - An estimate $\widehat{\mathbf{P}}$ of the observation probabilities (with $\widehat{p}_{ij}$ estimating $p_{ij}$, the probability of entry $(i,j)$ being observed).

7:       Compute normalized residuals for $(i,j) \in \mathcal{S} \cup \{(i_*, j_*)\}$,

$$
R_{ij}^{(m)} = \frac{|M_{ij} - \widehat{M}_{ij}^{(m)}|}{\widehat{s}_{ij}^{(m)}}.
$$

8:       Compute weights

$$
\widehat{w}_{ij} = \frac{\widehat{h}_{ij}}{\sum_{(i',j') \in \mathcal{S} \cup \{(i_*, j_*)\}} \widehat{h}_{i'j'}}, \qquad \widehat{h}_{ij} = \frac{1 - \widehat{p}_{ij}}{\widehat{p}_{ij}}, \qquad (i,j) \in \mathcal{S} \cup \{(i_*, j_*)\}.
$$

9:       Compute the weighted quantile

$$
\widehat{q}^{(m)}(i_*, j_*) = \text{Quantile}_{1-\alpha} \left( \sum_{(i,j) \in \mathcal{S}} \widehat{w}_{ij} \delta_{R_{ij}^{(m)}} + \widehat{w}_{i_* j_*} \delta_{+\infty} \right) \tag{21}
$$

10:    **end for**
11: **end for**
12: **Output**: $\left\{ \widehat{C}(i_*, j_*) = \left\{ m \in \mathcal{M} : R_{i_* j_*}^{(m)} \le \widehat{q}^{(m)}(i_*, j_*) \right\} : (i_*, j_*) \in \mathcal{S}^c \right\}$

---

We note that, when $\mathcal{M} = \mathbb{R}$ (or an infinite subset of $\mathbb{R}$), the computation of the prediction set is impossible in most of the cases. In that case, our algorithm can be modified via a trimmed or discretized approximation; these extensions are presented for the regression setting in the work of

Chen et al. (2016, 2018), and can be extended to the matrix completion setting in a straightforward way.

## C.3 Exact split conformalized matrix completion

In Algorithm 3, we present the exact split approach, which is less conservative than our one-shot approach given in Algorithm 1, but may be less computationally efficient. In this version of the algorithm, the quantile $\widehat{q} = \widehat{q}(i_*, j_*)$ needs to be computed for each missing entry since the weight vector $\widehat{w}$ depends on the value of $\widehat{p}_{i_*, j_*}$.

---

**Algorithm 3** `split-cmc`: split conformalized matrix completion

---

1: **Input**: target coverage level $1 - \alpha$; data splitting proportion $q \in (0, 1)$; observed entries $\mathbf{M}_{\mathcal{S}}$.

2: Split the data: draw $W_{ij} \overset{\text{i.i.d.}}{\sim} \text{Bern}(q)$, and define training and calibration sets,

$$\mathcal{S}_{\text{tr}} = \{(i, j) \in \mathcal{S} : W_{ij} = 1\}, \quad \mathcal{S}_{\text{cal}} = \{(i, j) \in \mathcal{S} : W_{ij} = 0\}.$$

3: Using the training data $\mathbf{M}_{\mathcal{S}_{\text{tr}}}$ indexed by $\mathcal{S}_{\text{tr}} \subseteq [d_1] \times [d_2]$, compute:

- An initial estimate $\widehat{\mathbf{M}}$ using any matrix completion algorithm (with $\widehat{M}_{ij}$ estimating the target $M_{ij}$);
- Optionally, a local uncertainty estimate $\widehat{\mathbf{s}}$ (with $\widehat{s}_{ij}$ estimating our relative uncertainty in the estimate $\widehat{M}_{ij}$), or otherwise set $\widehat{s}_{ij} \equiv 1$;
- An estimate $\widehat{\mathbf{P}}$ of the observation probabilities (with $\widehat{p}_{ij}$ estimating $p_{ij}$, the probability of entry $(i, j)$ being observed).

4: Compute normalized residuals on the calibration set,

$$R_{ij} = \frac{\left| M_{ij} - \widehat{M}_{ij} \right|}{\widehat{s}_{ij}}, \ (i, j) \in \mathcal{S}_{\text{cal}}.$$

5: Compute estimated odds ratios for the calibration set and test set,

$$\widehat{h}_{ij} = \frac{\widehat{p}_{ij}}{1 - \widehat{p}_{ij}}, \ (i, j) \in \mathcal{S}_{\text{cal}} \cup \mathcal{S}^c,$$

6: **for** $(i_*, j_*) \in \mathcal{S}^c$ **do**

7:     Compute weights for the calibration set and test point,

$$\widehat{w}_{ij} = \frac{\widehat{h}_{ij}}{\displaystyle\sum_{(i', j') \in \mathcal{S}_{\text{cal}}} \widehat{h}_{i'j'} + \widehat{h}_{i_* j_*}}, \ (i, j) \in \mathcal{S}_{\text{cal}}, \ \ \widehat{w}_{\text{test}} = \frac{\widehat{h}_{i_* j_*}}{\displaystyle\sum_{(i', j') \in \mathcal{S}_{\text{cal}}} \widehat{h}_{i'j'} + \widehat{h}_{i_* j_*}}.$$

8:     Compute threshold

$$\widehat{q}(i_*, j_*) = \text{Quantile}_{1-\alpha} \left( \sum_{(i, j) \in \mathcal{S}_{\text{cal}}} \widehat{w}_{ij} \cdot \delta_{R_{ij}} + \widehat{w}_{\text{test}} \cdot \delta_{+\infty} \right),$$

    where $\delta_t$ denotes the point mass at $t$.

9: **end for**

10: **Output**: confidence intervals

$$\widehat{C}(i_*, j_*) = \widehat{M}_{i_* j_*} \pm \widehat{q}(i_*, j_*) \cdot \widehat{s}_{i_* j_*}$$

for each unobserved entry $(i_*, j_*) \in \mathcal{S}^c$.

---

