# OpenReview forum: "Conformalized matrix completion"
_NeurIPS.cc/2023/Conference — NeurIPS 2023 poster_

### Official Review · Reviewer_u41g · 2023-07-04

**Soundness:** 3 good
**Presentation:** 3 good
**Contribution:** 2 fair
**Rating:** 6
**Confidence:** 4

**Summary:**

This paper addresses the problem of uncertainty quantification in matrix completion by developing a distribution-free method for predictive inference.  The authors propose a novel approach based on the conformal prediction framework, aiming to overcome the limitations imposed by stringent model assumptions such as low-rankness of the underlying matrix and the light-tailed noise.  Their method, referred to as Conformalized Matrix Completion (CMC; Algorithm 1), can be combined with any arbitrary matrix completion algorithm to provide confidence intervals for the imputed matrix entries.  Here, the “confidence” is measured with respect to the randomness inherent in the measurements of the matrix entries, assuming a randomized probabilistic measurement model.

To support their proposed method, the authors also present an intuitive explanation of the method and a theoretical guarantee.  They start by giving an intuitive exposition of how the (weighted) exchangeability plays a crucial role in constructing confidence intervals within the conformal prediction framework (Section 3.1).  Thereafter, they present Theorem 3.2, which establishes a theoretical guarantee on the average coverage of the confidence intervals, which serves as a key theoretical result of this paper.  Additionally, the authors illustrate their analysis by describing two examples of the missingness patterns (i.e., the random observation models) in Section 3.3.  Through this illustration, they explicitly evaluate the rate of the expected “weight gap” as expressed in Eq. (6), ultimately demonstrating satisfactory coverage of the constructed confidence intervals.

In Section 4, the authors report the results of their numerical simulations. These simulations serve two purposes: (1) comparing the performance of the proposed conformalized method against a model-based baseline (Sections 4.1.1 & 4.2), and (2) investigating the stability of the proposed method when the oracle knowledge of the measurement probability for each entries is not available (Section 4.1.2).

**Strengths:**

One of the primary strengths of this paper lies in its adaptation of the conformal prediction framework to address the challenge of uncertainty quantification in matrix completion. Unlike previous approaches in the literature, this method does not heavily rely on specific model assumptions. This combination of conformal prediction and matrix completion is a significant and original contribution, making the paper stand out in its field.

Another notable strength is the careful organization of the paper, which effectively presents the essential components of the study to support the authors' main claims. Section 2 provides a concise overview of the problem setup and evaluation metric. The proposed method is then described in detail, accompanied by an intuitive explanation based on the concept of exchangeability in Section 3.1. The main theoretical result is presented neatly in Section 3.2, and to further enhance the understanding, concrete examples of the missingness patterns are provided in Section 3.3. The paper is further strengthened by the inclusion of numerical study results in Section 4 and a comprehensive discussion in Section 5. As a result, the paper forms a clear and self-contained report of the study, ensuring that the readers can grasp the main ideas and findings with ease.

Overall, the adaptation of conformal prediction, the clear organization of the paper, and its ability to convey the study's main contributions effectively are key strengths of this research work.

**Weaknesses:**

While the paper exhibits notable strengths, there are a few areas that could benefit from further improvement and clarification. I have identified three main concerns that warrant attention.

Firstly, the authors frequently emphasize that their proposed method is "distribution-free" and does not rely on “any” assumptions about the underlying matrix (e.g., in line 60). However, it is important to explicitly acknowledge that the proposed method and analysis do depend on the probabilistic random observation model, which is necessary for the conformal prediction framework. While this limitation does not appear to be critical, it would be beneficial to provide a careful clarification of the assumptions and limitations to avoid potential confusion.

Secondly, although the authors highlight the limitations of existing uncertainty quantification approaches (lines 24 - 30), there is a lack of comparisons between the proposed method and these previous approaches. Consequently, it remains unclear whether and how the proposed method surpasses these existing methods. Including such comparisons would enhance the clarity and strengthen the argument for the superiority of the proposed approach.

Additionally, the issue of heavy-tailed noise is briefly mentioned by the authors (lines 32 - 33), who suggest that their method is less sensitive to the noise tail (lines 32-33, lines 257-261). However, this point is not adequately addressed beyond the comparison of the vanilla ALS method to the CMC-ALS method in Figure 1-(c), where the latter seems to be slightly better than the former, but still remains overly conservative; the difference in performance between the ALS and the CMC-ALS does not appear to be substantial (approximately 9x worse vs 6x worse than the oracle). To improve the cohesiveness of the paper, it would be beneficial to either elaborate further on this point or consider removing it if it does not significantly contribute to the main findings.

In summary, while the paper possesses strengths, it would benefit from addressing these weaknesses. Clearer clarification of assumptions and limitations, comparative analyses with existing methods, and a more comprehensive discussion of the performance in the presence of heavy-tailed noise would enhance the overall quality and cohesiveness of the paper.

**Questions:**

1.  I suggest the authors consider providing a clarification of the assumption on the measurement model, as mentioned in the first point of the "Weakness" section. This clarification would help readers better understand the specific assumptions and limitations of the proposed method.

2.  It would be valuable if the authors could compare their method with the approaches mentioned in the second paragraph (lines 24-30) by conducting simulations or other appropriate means. Such comparisons would enable a clearer assessment of how the proposed method outperforms or differs from existing methods.

3.  I am curious about the degree to which the proposed CMC method relies on the uncertainty estimate \hat{s}. Many matrix estimation algorithms only provide point estimates for the entries without accompanying uncertainty estimates. In Algorithm 1, when uncertainty estimates are unavailable, the authors set \hat{s}_{ij} = 1 by default (line 3). However, this default choice can be problematic. For instance, consider two scenarios: (1) estimating M and (2) estimating 10M. In scenario 2, the uncertainty should be ten times larger than in scenario 1, but the default choice cannot account for this. It would be beneficial if the authors could address this issue and discuss potential alternatives or adjustments to handle situations where uncertainty estimates are not provided.

4.  In Section 3.3, it might be helpful for readers to have either the upper bound for the expected weight gap or the resulting lower bound for average coverage presented as explicit corollaries.

5.  The caption of Figure 2 does not provide sufficient information to understand what is being compared. It would be helpful to revise the caption to clearly indicate the elements being compared or provide a brief description of the comparison being made.

**Limitations:**

While the authors outline potential future research directions, they do not explicitly address the limitations of their proposed method and approach. It would be beneficial if the authors could include a brief discussion of the technical limitations, such as the assumptions made in their approach. Additionally, providing insights into potential negative impacts when applying the method in real-world applications would further enhance the paper's practical relevance. However, it should be noted that as this paper primarily focuses on theoretical aspects, addressing these limitations in detail may not be deemed critical.

---

> ### Author Rebuttal · Authors · 2023-08-09
>
> Thank you for your helpful comments and feedback! Our replies below address the individual points raised in your review.
>
> - In response to the 1st Weakness (“Firstly, the authors frequently emphasize that their proposed method is "distribution-free" “),
> Thanks for the suggestions. We will add clarifications about the assumptions and limitations in the revision.
>
> - In response to the 2nd Weakness (“Secondly, although the authors highlight the limitations of existing uncertainty quantification approaches…“),
> In fact, in our experiment, we did compare our conformal methods with the existing uncertainty quantification approaches (lines 24 - 30). But we forgot to add references. In the updated manuscript, we will add corresponding references to existing approaches.
>
> - In response to the 3rd Weakness (“Additionally, the issue of heavy-tailed noise is briefly mentioned…“),
> Thanks for the suggestion on this heavy-tailed setting. In the appended pdf [Figure 1(l)], we further show the ratio between the obtained length and the oracle length for both als and cmc-als. Although the heavy-tailed problem is indeed a hard problem in terms of statistical inference, we can see that cmc-als actually corrects the conservativeness a lot over als. Moreover, in this heavy-tailed setting, we can see from the error bars that als is much more variable than cmc-als.
>
> - Question 1 is addressed in the response to the 1st Weakness, above.
> - Question 2 is addressed in the response to the 2nd Weakness, above.
> - In response to Question 3,
> Thank you for this question - this is an important point and we will be sure to explain more clearly in our revision. The values $\hat{s}\_{ij}$ capture the estimated relative, rather than absolute, noise levels across the different entries. Rescaling the entire matrix $\hat{s}$ by a constant (i.e., replacing $\hat{s}\_{ij}$ with $\hat{s}^\prime_{ij} = c\cdot \hat{s}\_{ij}$, for the same value c for all (i,j) does not change the outcome of the procedure; running with $\hat{s}^\prime\_{ij}$'s instead of $\hat{s}\_{ij}$'s would result in a quantile value $\hat{q}^\prime = \hat{q}/c$, and the resulting prediction intervals would therefore be identical. This means that, if we set $s_{ij}=1$ for all (i,j), this ensures that all the entries will be given prediction intervals of equal width, and can therefore be interpreted as assuming that the entries all have equal variance (but we are not assuming that the entries have variance equal to 1).
>
>
> - In response to Question 4,
> Thank you for the suggestion! The detailed result for Example 3.3.1 is shown in Section A.3.1 in the appendix and the detailed result for Example 3.3.2 is shown in Section A.3.2. To improve clarity, we will write them as explicit theorems and add pointers to them in the main text.
> - In response to Question 5,
> Thanks for the suggestions. We will add more details into the figure captions for Fig 1, 2,3

---

> > ### Comment · Reviewer_u41g · 2023-08-14
> >
> > I thank the authors for their efforts in providing a rebuttal and clarifications to address my concerns and questions. With confidence in the authors' commitment to further refining the manuscript in preparation of the camera-ready version, I reaffirm my initial moderately positive assessment.

---

### Official Review · Reviewer_1PHH · 2023-07-05

**Soundness:** 3 good
**Presentation:** 3 good
**Contribution:** 2 fair
**Rating:** 6
**Confidence:** 4

**Summary:**

This paper utilizes conformal inference techniques to address uncertainty quantification in the matrix completion problem. The authors present a novel method for constructing prediction sets for the missing entries estimated by any given matrix completion algorithm, employing a simple data hold-out strategy. The underlying assumption is that the data matrix $M$ is fixed, while randomness arises from the missingness of the matrix entries.

Specifically, each index is assumed to have a distinct probability of being observed, independent of the others. The authors demonstrate that when the missingness probabilities are uniform across all entries, the data become exchangeable, allowing for the direct application of standard conformal inference methods.

In cases with heterogeneous missing probabilities, the problem becomes more intricate and aligns with the conformal inference under the covariate-shift framework introduced by Tibshirani et al. (2019). To tackle this scenario, the authors propose applying the method developed by Tibshirani et al. (2019) and employ data-driven estimates of the missingness probabilities derived from parametric models.

The validity of the proposed method is established through both theoretical arguments leveraging existing results and empirical evaluation using synthetic and real data experiments. These experiments effectively demonstrate the efficacy and practical applicability of the proposed approach.

**Strengths:**

1)    This paper is both interesting and original, as it contributes to connection between conformal inference and matrix completion. The authors successfully present a reasonable missingness model and introduce a principled framework to apply weighted split conformal methods.

2)    The practical usefulness and potential impact of this paper are noteworthy. Until recently, uncertainty estimation in matrix completion was relatively unexplored, and prior methods were limited due to their heavy reliance on assumptions that often do not hold in practice. Therefore, the assumption-lean approach presented in this paper holds substantial promise and has the potential to significantly impact related fields.

3)    The paper is well-written, and the proposed method is clearly and effectively explained.

**Weaknesses:**

1)    Technical novelty and originality of theoretical contributions. While this paper seems to rely heavily on the results of Tibshirani et al. (2019) and Barber et al. (2022), the relationship between these works could be explained more clearly. In particular, it would be beneficial to clarify which aspects of the proofs are novel and which can be considered as special instances of prior work.

2)    Thoroughness of the numerical experiments. While the proposed method appears promising in theory, the numerical experiments lack convincing evidence. For example, in the four settings depicted in Figure 1, it would be desirable to observe under-coverage in $\texttt{als}$ when the signal-to-noise ratio is low or when the incoherence condition is violated, even with an oracle rank. However, both $\texttt{cmc}$ and $\texttt{als}$ exhibit similar performance when the rank is correctly chosen, with the advantages of $\texttt{cmc}$ primarily stemming from tuning the hypothesized rank. Further investigation and more comprehensive experiments would be useful to establish the illustrate of the proposed method.

3)    Realism of real-data experiments. The real data application may not provide significantly more information than a synthetic experiment since the authors begin with the full matrix $M$ and manually sample missing entries using a logistic missingness model. However, it remains unclear whether this logistic model accurately represents practical scenarios. Designing experiments based on more realistic missingness patterns would enhance the informativeness of the results. Additionally, it is expected that the benchmark $\texttt{als}$ would exhibit under-coverage in some heterogeneous settings, but the authors did not compare it to $\texttt{als}$. By neglecting this comparison, the practical relevance of the work is not made as clear as it could be.

4)    Limited empirical evaluation. The authors only consider the ``average coverage rate" as defined in Equation (3), which provides a relatively weak coverage guarantee. It would be helpful if the numerical experiments also included appropriate conditional coverage metrics to provide a more comprehensive evaluation.

**Questions:**

1)    Usefulness of the one-shot shortcut. The paper introduces a one-shot weighted conformal approach to reduce computational costs. However, it is important to consider potential issues with this relaxation. Firstly, the presence of extremely small probabilities $p_{ij}$ for some test points may lead to excessively large odd ratios $h_{ij}$, resulting in overly conservative predictions that may not be useful for other test points. Secondly, it is not entirely clear why the proposed algorithm without the one-shot relaxation would be prohibitively computationally expensive. Since the weight $w_{ij}$ has a simple form, it should not significantly increase evaluation time in practice.

2)    Improvement of technical notation. Some technical details require clarification. For example, in line 157 on page 5, the odd ratio is defined as $(1-p_{ij})/p_{ij}$, which poses a problem when $p_{ij}=0$ for some index $(i,j)$. Additionally, in the Supplement Material, lines 464-467, the variables $\mathbf{Z}$ and $\mathbf{W}$ are not explicitly defined, making it more difficult to verify the proof.

3)    Missing citation. In line 233-237, the benchmark prediction sets are mentioned, which are believed to follow from the asymptotic results in Chen et al. (2019). However, the citation for this reference is missing. Including the appropriate citation will provide proper attribution and give readers an opportunity to explore the referenced work.

4)    Consideration of alternative methods. There are other matrix completion algorithms available that provide uncertainty quantification, such as matrix completion with Gaussian Copula (Zhao et al, 2020). It would be beneficial for the authors to implement additional benchmarks using these alternative methods to strengthen the validity and robustness of the data experiments.

**Limitations:**

The main limitations of this paper concern its technical novelty compared to prior work on conformal inference and the depth of its empirical evaluations.

---

> ### Author Rebuttal · Authors · 2023-08-09
>
> Thank you for your helpful comments and feedback! Our replies address the individual points raised in your review.
> - [Weakness 1] Thank you for the question. The main novelty of our work is in reformulating the matrix completion problem as an instance of weighted exchangeability, for which weighted conformal prediction (WCP) can be applied (Lemma 3.1). Secondly, many existing theoretical results using WCP either assume that the true weights w are known, or require ||\hat{w}-w|| to be small with high probability; this can be quite loose, and we have been able to work with an error term that measures only the expected value of the gap. We will explain the novelty and contributions more clearly in our revision.
> - [Weakness 2] We set the hypothesized rank r=8 in these two settings. [Figure 1(j)] We modify Setting 2 (line 243) with d=400 and p=0.1. The coverage rate of als drops to 0.74 while cmc-als has the desired guarantee. [Figure 1(k)] We modify Setting 4 (line 246) by replacing t_{1.2} with Cauchy(0,1) for the entries. als does not have coverage guarantee while the coverage of cmc-als is exact.
> - [Weakness 3] Real data: In the pdf [Figure 1(e)], we consider the sales dataset where the missingness occurs in the way that: For weekdays, each entry is observed with pr=0.8. On weekends, as the stores are likely to be operated by less experienced interns or are less frequent to report the sales data, each entry is observed with a lower probability, e.g. 0.8/3. Moreover, as there could be a subgroup of stores that are less frequent in sales reporting, 200 stores are randomly sampled, for which pr=0.8/3. We use the logistic model with k=5 to estimate p_{ij}.
> In the existing literature with theoretical validity guarantee, the implementation of als is not designed for heterogeneous sampling. To compare with als, we estimate p_{ij} via the uniform sampling model.
> In the pdf [Figure 1(h)], we use the setting same as Figure 2(f) with hypothesized r=8 and observe a coverage gap of 1% for als. When the noise is adversarial [Figure 1(i)], the cmc-als has a coverage gap of 1% due to the estimation error of w while the coverage of als drops to 0.86.
> - [Weakness 4] In the pdf [Figure 1(f)], we evaluate the local performance of cmc-als when conditioning on (p_{ij} = p_0), i.e. a subpopulation determined by the value of sampling probability. We use the setting same as Figure 2(b) with the hypothesized r=12 and p_0 ranging from 0.1 to 0.9 with a step size of 0.1. The conditional coverage is approximated via kernel smoothing: with the indicators A_{ij} for coverage, we use the weight K_{ij} = \phi_{p_0,h}(p_{ij}) and calculate the conditional coverage by (\sum A_{ij} K_{ij})/(\sum K_{ij}). Here \phi_{\mu,\sigma} is the density function of \calN(\mu,\sigma^2) and h=0.05.
> When p_0 increases from 0.2 to 0.8, the conditional coverage increases and stays around 0.9. At p_0=0.1 and 0.9, the coverage varies much, which is due to the small effective sample size for the edge values of p_{ij}. Moreover, for uniform sampling, since the rows & columns are generated i.i.d., there are no meaningful subgroups of the data to condition on.
> - [Question 1] Thanks for this question – we realize now that our explanation in the paper was not clear. While there is a small computational benefit, indeed as the reviewer points out, it is not really significant. The real benefit is that the one-shot method produces a much more interpretable answer: the final output of the method is of the form on line 168 for a single value \hat{q} – that is, the same scaling or inflation factor is applied across the entire matrix.
> We would like to pause to clarify why we would NOT want to have different rescaling factors \hat{q}_{ij} at different entries. In fact, it may appear initially that this output would be more meaningful – we might have different levels of uncertainty at different locations (i,j), so would it not be beneficial to choose \hat{q}_{ij} adapting to the inflation of an entry (i,j)?
> The answer to this is that, without the one-shot simplification, the different values of \hat{q}_{ij} are actually unrelated to the issue of higher or lower uncertainty about our estimates of \hat{M} and \hat{s} at this entry. They only have to do with differences in the weights vectors - an entry (i,j) would have a higher \hat{q}_{ij} based solely on its (estimated) weight \hat{h}_{ij} that reflects its probability of being sampled. Thus, the \hat{q}_{ij}'s are not actually reflecting a meaningful notion of local noise/uncertainty and in our opinion the method is more interpretable with the simplification.
> With that said, of course we do not want to propose a method that is needlessly conservative. To that end, we have carried out a simulation to verify that the two versions give essentially the same performance in the pdf [Figure 1(g)]. The setting is the same as Figure 2(b) with hypothesized r=12.
> - [Question 2] Thanks for pointing this out. We assume that the p_{ij}'s are nonzero in order for WCP to be well defined. We apologize for omitting to state this explicitly and will add this assumption to our revision.
> Notations Z & W – we apologize for the confusing notations in the appendix. The matrix W consists of entries W_{ij} defined on line 2 in Algorithm 1. The matrix Z consists of Z_{ij} which are the indicators of non-missingness, i.e. Z_{ij} = 1{(i,j) is observed}.
> - [Question 3] The reviewer is correct. For model-based inferential results, we use results from Chen et al. (2019). We will add this reference in place.
> - [Question 4] Thanks for this reference. This paper gives theoretical guarantees under the Gaussian copula assumption for the data, and thus does not provide an alternative mechanism for distribution-free theory. We will add the paper to our discussion. Unfortunately, due to space constraints, we cannot add an additional experiment to compare, but we expect to see a loss of coverage when the model assumption is strongly violated.

---

> > ### Comment · Reviewer_1PHH · 2023-08-14
> >
> > I appreciate your responses to my inquiries! The primary concerns that were causing confusion for me have been effectively resolved. While I continue to hold some reservations about the extent of technical innovation, which has somewhat constrained my confidence in assigning an exceedingly high rating, I acknowledge that the paper is accurate, intriguing, and holds practical value. As a result, I've decided to adjust my score from 5 to 6. Thank you!

---

### Official Review · Reviewer_1TvL · 2023-07-06

**Soundness:** 2 fair
**Presentation:** 3 good
**Contribution:** 3 good
**Rating:** 5
**Confidence:** 2

**Summary:**

This paper presents a distribution-free method for constructing prediction intervals in the matrix completion problem, where randomness only arises from the sampling of observed entries. The approach utilizes weighted conformal prediction and establishes a lower bound on the probability of each unobserved entry being included in the prediction interval. The lower bound relies on the estimation error of sampling probabilities, which can be negligible if accurately estimated.

**Strengths:**

* Overall, the paper is clearly written.
* The utilization of weighted conformal prediction in the matrix completion problem is quite novel.
* The study shows that the resulting prediction interval performs well with well-estimated sampling probabilities.

**Weaknesses:**

* Incorrect estimation of sampling probabilities may significantly degrade the quality of the prediction interval.

**Questions:**

* Are there any experimental results when the sampling probabilities are completely misspecified, e.g., when the heterogeneous missingness is misspecified as uniform sampling?
* I'm curious about the impact of the estimated values of $\hat{s_{ij}}$'s. Does misspecification of these values have a substantial negative impact on the results? Are there any empirical findings regarding this?
* Is there a rationale for the estimation of theta at Line 237? If there is a reference concerning this, could you provide it?
* Since the constructed prediction interval is for M, rather than M*, wouldn't it be appropriate to employ an approach for approximately low-rank matrices in experiments? With an approximately low-rank matrix completion approach, how does the estimation of theta change?
* What does Line 277 mean?

**Limitations:**

Yes.

---

> ### Author Rebuttal · Authors · 2023-08-09
>
> Thank you for your helpful comments and feedback! Our replies below address the individual points raised in your review.
>
> - In response to the Weakness and to the 1st bullet point under Questions (“Are there any experimental results when the sampling probabilities are completely misspecified…”),
> In the attached pdf, we conducted simulations in the following 4 settings:
>
>     - Setting [a] [Figure 1(a)] The underlying missingness follows the rank-one model where $p_{ij} = a_i b_j$, both $a_i$ and $b_j$ are generated i.i.d. from Unif(0.2,1). The noise is adversarial (line 273). But we estimate p_{ij} via the one-bit matrix completion based on the logistic model (working model with hypothesized rank k=5).
>
>     - Setting [b] [Figure 1(b)] The underlying missingness follows the logistic model as in Example 3.3.2 with $k^*=5$ and we adopt the adversarial noise on line 273. But $p_{ij}$ is estimated under the assumption of uniform sampling (working model), i.e. $\hat{p}_{ij}$ = $\hat{p}$ (line 239).
>
>     - Setting [c] [Figure 1(c)] Same as Setting [a] except that we use the random noise on line 276.
>
>     - Setting [d] [Figure 1(d)] Same as Setting [b] except that we use the random noise on line 276.
>
> 	From the results, we can see that when the noise is generated following line 276 (i.e., random noise model), where the values of entries are independent of the sampling probabilities, the misspecification of the sampling model only slightly affects the coverage. Moreover, when the noise is generated in the adversarial way, where the values of entries depend on $p_{ij}$’s, we can see that the coverage with a misspecified sampling model is lower than the target level, but is above 0.85 in practice, which depends on the divergence between the true and the working sampling models. We will add this experiment to the appendix.
>
> - In response to the 2nd bullet point under Questions
> In fact, $\hat{s}_{ij}$ doesn’t affect the validity of the prediction intervals. Similar to the estimate $\hat{M}$, they are both initial estimates. Regardless of their accuracy, our conformal methods help correct them and offer provable predictive coverage.
>
> - In response to the 3rd bullet point under Questions (“Is there a rationale for the estimation of theta at Line 237?”):
> Eq [19] in the paper “Inference and uncertainty quantification for noisy matrix completion” published in PNAS provides the rationale for this $\hat{\theta}$. In words, it can be viewed as the asymptotic variance of the model-based estimate when the model is correctly specified.
>
> - In response to the 4th bullet point under Questions (“Since the constructed prediction interval is for $M$, rather than $M^*$...”):
> When we use model-based estimate ALS, we used the prediction intervals for M (i.e., low-rank matrix plus noise), instead of confidence intervals for $M^*$. So our comparisons are fair in this sense.
>
> - In response to the 5th bullet point under Questions (“What does Line 277 mean?”)
> We apologize for any confusion. We will reword this in the revision. To clarify:
> The matrix ($p_{ij}$) is drawn by setting $log[ p_{ij}/(1-p_{ij}) ] = \langle a_i,b_j \rangle$ where vectors $a_i$, $b_j$ are drawn as in lines 264–266. To draw the sigma’s, we can think of first drawing an independent copy of p – say, ($p^\prime_{ij}$) – and then setting $\sigma_{ij} = 1/2p^\prime_{ij}$ for each (i, j). Equivalently, we are drawing new $a_i$, $b_j$ vectors, and then defining $\sigma_{ij}$ by taking $log[(1/2\sigma_{ij}) / (1-1/2\sigma_{ij}) ] = \langle a_i,b_j\rangle$.

---

### Official Review · Reviewer_bWVG · 2023-07-07

**Soundness:** 3 good
**Presentation:** 2 fair
**Contribution:** 2 fair
**Rating:** 5
**Confidence:** 2

**Summary:**

This paper proposes to use conformal prediction for uncertainty quantification of matrix completion. The proposed conformalized matrix completion offers provable predictive coverage regardless of the accuracy of the low-rank model. Empirical results on simulated and real data demonstrate that cmc is robust to model misspecification while matching the performance of existing model-based methods when the model is correct.

**Strengths:**

Strength 1. This paper studies conformal prediction for matric completion with interesting theoretical and algorithmatical findings.
1.1） Through conformal prediction, this paper proposes distribution-free confidence intervals for the completed entries in matrix completion. Benefiting from the use of conformal prediction, the validity is free of any assumption on the underlying matrixs and holds regardless of the choice of estimation algorithms. By proving the (weighted) exchangeabiity of unobserved and observed units when they are (non-uniformly) sampled without replacement from a finite population.
1.2） A provable lower bound for the coverage rate is provided when the sampling mechanism is unknown.
1.3） A one-shot conformalized matrix completion approach is proposed for higher computational efficiency.

Strength 2. Experiments on both synthetic and real data suggest the effectivenss of the proposal.

**Weaknesses:**

Weakness 1: One weakness of the proposed uncertainty quantification method for model calibration is that the probability bounds it provides may not be very precise for specific observation models and optimization algorithms. While the method is advantageous in that it is not limited to any particular model or algorithm, there is a possibility that the uncertainty estimates can be too broad for certain scenarios. This potential issue should be acknowledged and addressed in the research.

Weakness 2: Another weakness is that the proposed approach does not assume any specific structure on the underlying matrix. As a result, it may not effectively utilize any existing structure in the matrix. To overcome this limitation, the proposed method can be extended to vector completion problems, which could exploit the matrix structure more efficiently. The authors should consider discussing this point and its potential implications.

Weakness 3: The paper could benefit from improved clarity in its explanations. It would be more reader-friendly if the authors provided more intuitive explanations for the newly introduced quantities, such as the quantity labeled as (6) in the paper. Enhancing clarity in the presentation of the research would greatly improve its accessibility to readers.


**Questions:**

Please refer to the "Weaknesses".


**Limitations:**

The limitations have not been adequately addressed in the main text.

---

> ### Author Rebuttal · Authors · 2023-08-09
>
> Thank you for your helpful comments and feedback! Our replies below address the individual points raised in your review.
>
> - In response to Weakness 1 and Weakness 2,
> Thank you for these points. These questions are inherent to the conformal prediction framework, and are not specific to our specific method or to the specific setting of matrix completion. Our replies are therefore general, and apply to any implementation of the conformal framework to any prediction problem:
>
>     - To reply to Weakness 1, the conformal prediction method is designed to provide uncertainty quantification to ANY base estimation algorithm – it is indeed true that the resulting intervals may be very wide if the base algorithm is a poor estimate, but this is not a drawback of conformal. A useful analogy is using cross validation to estimate the error of a model. If we fit a model that actually has very high out-of-sample error, and then cross validation estimates a very high risk for this model, that is a success of CV (i.e., we have correctly identified that this model does not generalize well), not a failure. Analogously, if conformal prediction is applied to a poor estimation model and consequently gives a very wide prediction interval, that is a success of conformal (i.e., we have correctly identified that, when building a valid prediction interval around this particular \hat{M}, we need to make it very wide if we want to achieve coverage at a certain level), not a failure. This view of conformal can be captured by describing conformal as a “wrapper method” – its job is to provide guard rails around any base algorithm (i.e., any procedure that we use to produce \hat{M}), and it’s up to the analyst to choose a good base algorithm.
>
>     - To turn to Weakness 2, now we can see that actually the nature of conformal as a wrapper method, addresses this concern from the referee. While conformal itself does not use any model or property for the matrix, the base algorithm that produces \hat{M} is free to use ANY prior knowledge about structure in the matrix (and we do not need to worry if our assumptions are exactly correct!). For example, in our implementation of cmc-als, the base algorithm to produce \hat{M} is the als algorithm, which (implicitly) assumes that M follows a signal-plus-noise structure, M* + (iid noise), with incoherent and low rank M*. If this assumption is correct or approximately correct, our estimator \hat{M} will be excellent and conformal will provide a precise prediction interval – this is why cmc-als is competitive with als in terms of interval width, in the setting where the assumptions of the model hold. The benefit of cmc (and conformal in general) is that it maintains validity even if the assumptions of the base algorithm are wrong (as we see in our simulations for misspecified settings). We did not discuss these ideas extensively in the paper because this interpretation of the conformal framework is broadly applicable to all the conformal prediction literature and is not specific to our setting. However, we will add a bit more about these ideas into our revised paper to give more context to readers who are less familiar with the conformal prediction framework.
>
>
> - In response to Weakness 3,
> Thanks for the suggestions on presentation. Lines 179 - 182 explain intuitions for the newly defined $\Delta$. In words, $\Delta$ quantifies the estimation error in $\hat{w}$ w.r.t.~the oracle weight $w^*$. The smaller $\Delta$, the better the estimation of the weights, and hence the smaller the coverage gap from the nominal level $1-\alpha$.
> We will go through the paper and add more explanations for other quantities to improve clarity.

---

> > ### Comment · Reviewer_bWVG · 2023-08-14
> > **Response to the reply**
> >
> > Thanks for the clarification for my concerns.
> >
> > - My first two questions are inherent to the conformal prediction framework, and are not specific to our specific method or to the specific setting of matrix completion. I think the authors' response is reasonable.
> >
> > - The clarification about $\delta$ makes sense.

---

### Author Rebuttal · Authors · 2023-08-09

We are grateful to all the reviewers for their helpful feedback, comments, and suggestions on our manuscript. In the comments below, we have replied to each reviewer’s points individually.
The attached pdf contains additional simulation results and details of each setting are stated in the rebuttal.

---

### Decision · Program_Chairs · 2023-09-21

**Decision:**

Accept (poster)

**Comment:**

The reviews seem high quality and agree that this is a nice contribution that advances the literature on matrix completion using ideas from conformal inference to establish confidence bounds on imputations.